# Trends in Chemical Wood Surface Improvements and Modifications: A Review of the Last Five Years

**Pierre Blanchet** and **Simon Pepin** *

Natural Sciences and Engineering Research Council of Canada (NSERC) Industrial Chair on Ecoresponsible Wood Construction, Université Laval, 2425 Rue de l'Université, Québec, QC G1V 0A6, Canada; Pierre.Blanchet@sbf.ulaval.ca
* Correspondence: simon.pepin.1@ulaval.ca

**Abstract:** Increasing the use of wood in buildings is regarded by many as a key solution to tackle climate change. For this reason, a lot of research is carried out to develop new and innovative wood surface improvements and make wood more appealing through features such as increased durability, fire-retardancy, superhydrophobicity, and self-healing. However, in order to have a positive impact on the society, these surface improvements must be applied in real buildings. In this review, the last five years of research in the domain of wood surface improvements and modifications is first presented by sorting the latest innovations into different trends. Afterward, these trends are correlated to specifications representing different normative, ecologic and economic factors which must be considered when expecting to introduce a wood treatment to the market. With this review, the authors hope to help researchers to take into consideration the different factors influencing whether new innovations can leave the research laboratory or not, and thereby facilitate the introduction of new wood surface treatments in the society.

**Keywords:** wood; surface; improvements; modifications; treatments; coatings; plasma modification; surface impregnation





## 1. Introduction

Although wood is already a well-appreciated construction material, it is of primary importance to properly protect it and reach its full potential on the market. Indeed, it is crucial that the use of wood in buildings increases, as it is a renewable and bio-based material that can store large amounts of carbon dioxide and substitute for less eco-friendly materials such as steel and concrete [1,2]. For that matter, the use of wood in construction was even designated as a key tool to fight climate change by the FAO [3]. In order to achieve this goal, efforts must be deployed to improve attitudes of the public and architects toward its presence in buildings.

A central issue with wood is its combustibility, which is a subject of high legislative importance and a major source of concerns for the public. Wood can however be exposed to other sources of degradation according to its use (indoor, outdoor with or without exposition to rainfalls, ground contact, marine environment, etc.) [4]. Wood exposed outdoor can be weathered by numerous abiotic elements such as wind, dust, rain, and sunlight. Both air moisture and liquid water can be absorbed by the wood's hygroscopic nature and produce dimensional changes, which will lead it to cup, warp and crack [5]. If suitable conditions are met, living organisms such as decay fungi, molds, insects, and marine borers may feed on its structural components or on the nutrients contained in its parenchyma [4]. All of these will inevitably modify its chemical and physical properties, as well as its physical appearance, which will reduce its lifespan. This reality is of high ecological and economical importance, as defective wood pieces need to be replaced and make wood less competitive against other materials.

Good building practices can prevent the degradation of wood by reducing its exposition to some sources of degradation. Choosing high quality materials, such as naturally durable species or heartwood, can also help to extend the service life of wood products. However, the price and availability of durable species, as well as the increasing importance of fast-growing, lower quality lumber call for other alternatives [6]. In order to make wood more performant toward degradation, wood treatments become a very reliable solution. Many treatments have been developed over the last decades, including thermal and chemical modifications [7–10], thermo-mechanical densification [11,12], and impregnation [13–15]. Wood can be impregnated through vacuum/pressure methods to introduce biocidal, hydrophobic, and fire-retardant materials deeply in its structure [16,17]. Finally, wood can simply be protected on its surface with a variety of organic, mineral, or metallic coatings [18,19], or by different surface treatments.

Coatings and surface treatments can have different functions, from protecting wood from degradation to imbuing it with new and attractive properties. A major aspect of the research of wood surface protection is hydrophobization, since water is omnipresent and it can cause many kinds of defects, from cracking to promoting fungal growth [5]. A material is considered hydrophobic when the contact angle of water on its surface is above 90°, and superhydrophobic when its contact angle is over 150° and its roll-off angle (the tilt required for the droplet to slide) is below 10° [20]. While hydrophobicity is fairly easy to achieve, superhydrophobicity requires a rough and carefully designed micro-/nanoscale architecture and a very low surface free energy. This property is hard to maintain over a long time, as the micro-/nanoscale structure is quite fragile. Self-healing properties can be quite useful to maintain superhydrophobicity, as it can repair the micro-/nanoscale structure after physical damages such as cutting and abrasion [21]. Self-healing can be achieved in two ways, either by having a reserve of mobile hydrophobic materials to fill the damages or by reestablishing the initial structure through stimuli. A great side effect of superhydrophobicity is that it promotes other interesting surface properties such as anti-sludging and self-cleaning [22]. These properties rise from the fact that contaminants dissolved in water drops will hardly be left on surfaces on which water cannot adhere, and that water-soluble contaminants will be washed off when water slides away. Another way to obtain a self-cleaning surface is by improving its photocatalytic properties, allowing it to decompose organic compounds.

Photostability is another important property of wood coatings and surface modification. Ultraviolet rays from the sun promote the weathering of wood by degrading the polymeric constituents of its surface, mostly lignin [23]. This phenomenon results in a pale-looking wood with a decreased adhesion and can lead to the loss of its coating, which would expose wood to more sources of degradation [24]. Opaque and semi-transparent coatings are less prone to photodegradation, as their pigments readily absorb the UV rays. However, if a transparent coating is used in order to appreciate the natural aesthetics of wood, then UV-blockers are needed to prevent its weathering [25].

As mentioned earlier, fire protection is a central stake in wood protection [26]. The protection of wood from fire includes delaying the initial burning time, reducing the amount of smoke and heat released upon burning, and slowing down the spread of the flames [27]. It is often assured by impregnation with fire retarding materials, but it can also be achieved through surface protection. Some of the methods to reach this objective include inorganic coatings [28], modified organic coatings and fire-retardant fillers [19]. Other important characteristics can be brought about by surface treatment such as improved mechanical properties (hardness, abrasion resistance), chemical stability, and resistance to biodegradation.

As presented by Dimitrakellis and Gogolides, numerous methods exist to coat and modify the surface of wood, many of which can be divided in two groups: bottom-up and top-down [20]. Bottom-up methods involve the gradual building of the surface protection and include methods such as layer-by-layer deposition and chemical vapor deposition. Conversely, top-down methods shape a bulk of material, or wood itself, into the desired

end-product; such approaches include plasma modification. Commonly, surface coatings and modifications will rely on techniques such as dipping, spraying, brushing, chemical grafting, electroless deposition, chemical plating, and hydrothermal deposition.

The wide variety of treatment methods and chemicals used in wood-surface protection is indeed of great scientific interest and is needed to find solutions suitable for the different uses of wood. However, the large amount of knowledge produced on the subject can be confusing to readers trying to plan future research, while looking for the most promising treatments for a given application. In this review, a summary of the last five years of research in the domain of wood surface improvements and modifications will first be presented in order to establish the major trends. Subsequently, these trends will be analyzed from different perspectives, including some normative, environmental, and economical aspects associated with wood protection, in order to see which ones perform the best in each category. The aim of this review is not to decide which treatments are worthwhile and which ones should be abandoned, but rather to have a reflection on the strengths and weaknesses of the current research trends while discussing some key elements associated with the deployment of new wood protection technologies in the industry.

## 2. Methodology

The object of this publication is to review the last five years of literature in wood surface improvements and modifications, ranging from 2016 to 2020. To give an accurate description of the very latest trends, the publications up to 20 May 2021, were also included in the review.

To find as many relevant publications as possible, complementary methods were used to investigate the literature. First, a systematic approach was adopted to screen five databases (Compendex, Inspec, GEOBASE, GeoRef, and Knovel) on Engineering Village. Then, a research strategy based on keywords was carried out on Google Scholar, while publications and reviews were screened to find additional publications.

In order to keep the number of publications to a reasonable level and to allow good comparability between the treatments, strict exclusion criteria were chosen (Table 1). As a general statement, only treatments performed on solid wood were considered. In this way, the performances of the wood-surface treatments are not affected by gluing, pre-treatments, and transformations, allowing for a better comparison between the different studies. Likewise, treatments performed on wood-based materials such as panels, wood-plastic composites, and transparent wood were excluded. Also, treatments reaching deeper than the first few layers of cells of the wood specimens, such as acetylation, thermo-modification, thermo-mechanical densification, and transparent wood were excluded, as they cannot be considered surface treatments. Finally, the review focuses principally on wood used in construction; as such, treatments aimed at other uses such as water-oil separation or wooden artifacts restoration were also excluded to keep common thread between the publications.

As a result of searching in five databases at the same time, the number of publications found on Engineering Village included a lot of duplicates and, consequently, was very high. After removing the duplicates from multiple databases and between the different queries, 745 publications remained. The titles of the publications were first screened for relevance to the subject of the review; many titles specified containing excluded content (thermo-modified wood, pressure impregnation, etc.) or were simply irrelevant, leading to the rejection of 522 studies. The abstract and, when needed, the core of the 223 remaining publications were then screened for eligibility, of which 90 were included in the review. The research on Google Scholar and the publications cited in other texts led to a total of 803 publications; after removing the duplicates and the publications already found in the systematic research, 376 new publications were screened, 122 of which were included in the review. The keywords of the different queries are detailed in Table 2 and the PRISMA flow diagram is presented in Figure 1.

The 212 publications found in the literature were divided in three categories: coatings, surface modification and surface impregnation. Coatings were the most studied method for the protection of wood surfaces, with 144 publications. A total of 52 publications were included in the surface modifications and the last 16 described surface impregnation. The results were not compared between the studies, as the methods employed to test the properties of wood surfaces differed a lot between the studies.

**Table 1.** Justifications for the exclusion criteria.

| Exclusion Criteria | Reason |
|---|---|
| Thermo-modified wood<br>Chemically modified wood<br>Thermo-mechanically densified wood<br>Transparent wood<br><br>Glued wood product | These treatments were excluded from the review as they affect the treated wood deeply below the surface. Also, treatments performed on pre-modified wood specimens were excluded as their chemical and mechanical properties may be different and the performances of the treatments may not be comparable with treatments on solid wood. Glued wood products (glulam, plywood, OSB, etc.) were excluded as their mechanical properties and permeability may be affected. |
| Impregnated wood | Wood products impregnated before the surface treatment were not considered, as their chemical properties may be altered. |
| Deep impregnation | Only surface impregnation was relevant to the review; methods allowing deep longitudinal or radial impregnation were rejected. |
| Treatments unrelated to construction timber | To keep a certain logic between the publications presented, only the papers related to timber treatments were included. |

**Table 2.** Keywords of the queries on Engineering Village and Google Scholar.

| Engineering Village | Google |
|---|---|
| (1)-Wood AND surface AND (modification OR treatment) | (a)-Wood AND surface<br>(b)-(a) AND improvement |
| | (c)-(a) AND properties |
| | (d)-(a) AND modification |
| (2)-(1) AND (coatings OR impregnation OR plasma OR nanotechnology) | (e)-(a) AND treatment |
| | (f)-(d) AND treatment |
| | (g)-(a) AND impregnation |
| | (h)-(a) AND plasma |
| | (i)-(h) AND treatment |
| (3)-Wood AND surface AND (fire OR hydrophocity OR hydrophilicity OR uv OR self-healing OR hardness OR abrasion OR biodegradation) | (j)-(h) AND modification |
| | (k)-(g) AND treatment |
| | (l)-(g) AND modification |
| | (m)-(l) AND treatment |
| | (n)-(a) AND nanotech* |
| (4)-(2) And (fire OR hydrophocity OR hydrophilicity OR uv OR self-healing OR hardness OR abrasion OR biodegradation) | (o)-Wood AND (coatings OR finishes) |
| | (p)-(a) AND chemical AND densification |
| | (q)-(a) AND (coatings OR finishes) |
| | (r)-Wood AND self-healing |
| | (s)-(o) AND nanotech* |
| | (t)-(a) AND (Modification OR Plasma OR coating OR finishes OR chemical) AND (treatment OR impregnation OR nanotech* OR densification |

Nanotech* was used for the truncation of words starting with nanotech.

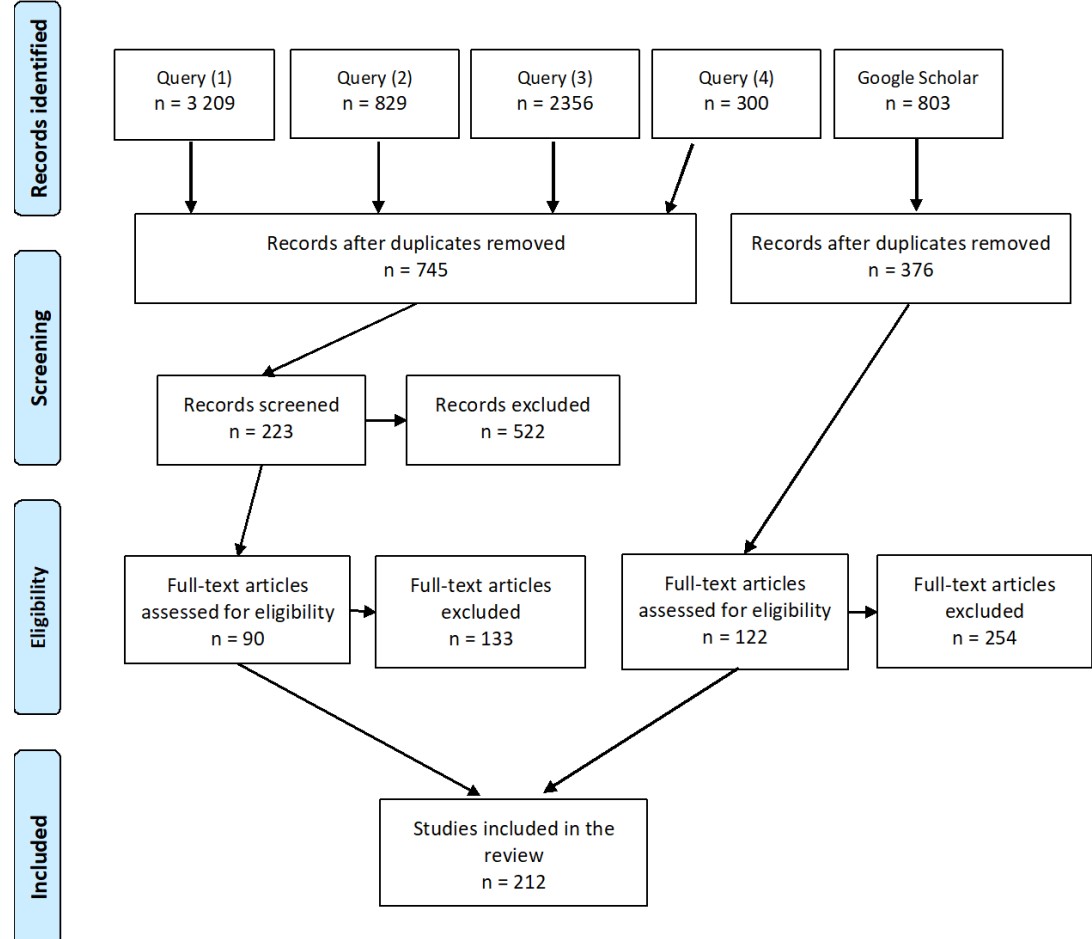

**Figure 1.** PRISMA flow diagram for the two searches strategies [29].

### 3. Trends of the Last Five Years

In this section, the publications covering the last five years of research in the domains of wood surface improvement and modification are presented. They are grouped in three main categories, namely coatings, surface modification and surface impregnation; because of the large number of papers published on the coatings and surface modifications, these are further divided into a few subcategories. Thereafter, the publications in each subcategory are sorted into different trends. Due to the large number of publications presented into the review, only some publications in each trend are detailed, while the other publications are simply presented. The detailed publications are usually chosen to represent the different properties obtained into each trend, because they performed particularly well, or because they present interesting features (bio-based materials, very high durability, etc.). Although extensive characterization of the surfaces and coatings is usually carried out in the original publications, this review focuses more specifically on the practical performances of the surface treatments (hydrophobicity, thermal stability, photostability, durability, etc.) in order to keep this section at a reasonable length. However, readers are encouraged to read the full publications to learn more about important elements such as transparency, gloss, roughness, chemical composition, and surface morphology.

#### 3.1. Coatings

Coatings were the most intensively studied methods for the protection of wood surfaces during the last five years. A total of 144 papers were found on this subject, using diverse materials and application methods to improve surface properties, such as

hydrophobicity, hardness, abrasion resistance, photostability, thermal stability, self-healing, self-cleaning and more.

### 3.1.1. Organic Coatings

Organic coatings are very common in wood protection, including waxes, oils, and film-forming resins such as acrylics, alkyds and polyurethanes. While they can be improved with additives, a popular approach found in the latest literature to enhance the properties of film-forming organic coatings was to use strategic reactives to change key properties of the resin itself. Different fire-retardant coatings were reported using phosphorus containing monomers or reactive diluents [30–33]. Lokhande et al. used glycidyl methacrylate, piperazine, and cyclic ethylene chlorophosphite to develop a diacrylate reactive diluent yielding UV-cured coatings with increased thermal resistance, hardness, hydrophobicity and stain resistance [34]. They found that the thermal properties of the coating would improve with the content of their reactive diluent, a concentration of 25% increasing the weight-loss temperatures, the heat-index resistance (from 149.7 °C to 184.0 °C), the practical char yield at 600 °C (7.12% to 22.21%), and the limiting-oxygen index (23% to 33%) when compared to the same coating without the reactive diluent. A similar coating was developed by Mulge et al. using epoxy acrylate oligomers and phenylphosphonic dichloride [35]. However, it was found that while a higher P content would improve the thermal stability of the coating, it would eventually impair its physical properties. In a similar way, Paquet et al. created self-healing, film-forming coatings by using acrylic monomers and oligomers containing many hydroxyl groups [36]. By using 2-hydroxyethyl methacrylate and an aliphatic urethane acrylate oligomer (Ebecryl 4738), the obtained polymer that could completely heal a 5 μm deep scratch or regain 83% of the gloss lost to abrasion after being heated to 80 °C for two hours. Another use for this practice was to build fast UV-curing acylic [37] and polyurethane-acrylate [38] formulations.

The same strategy was used to prepare more environmentally friendly coatings by using bio-based materials as a reactive. Raychura et al. prepared polyurethane coatings by reacting diisocyanates with fatty amides of mahua [39] and peanut [40] oils. The obtained wood coatings scored 100% on a cross-cut adhesion test, 1H or 2H on pencil hardness tests, and had a good stability to weak acids, water and NaCl solutions. Interestingly, wood coatings with good resistance to termites and/or white rot fungi could be obtained by reacting the starch from a yam (*Dioscorea hispida* sp.) with polyvinyl alcohol [41] or polyacrylamide [42]. Another antiseptic coating was prepared by Dixit et al. with citric acid and glycidyl methacrylate [43], which had high adhesion (5B), pencil hardness (6H), solvent resistance and a 15 mm zone of inhibition against the bacteria *Staphylococcus aureus*. It will be shown later in the review that cellulose nanocrystals (CNCs) were extensively studied as additives for organic coatings; Kong et al. [44], however, innovated by using CNCs as a reactive to imbue a waterborne polyurethane coating with a higher hardness and resistance to abrasion. They found that using only 0.1% of CNCs modified with 2,2,6,6-tetramethylpiperidine-1-oxyl (TEMPO) in their coating would increase its tensile strength and tensile elongation by 59% and 55%, respectively. More bio-based chemicals were studied by other workers, including soybean oil [45], rapeseed oil [46], and biosourced alcohols and acids [47].

Some organic coatings were prepared using very distinctive methods and could not be categorized into a specific trend, but still deserve some attention [48–50]. Janesch et al. dip-coated spruce in tung oil, bee wax, or a mix of both, before sifting sodium chloride (NaCl) on the freshly coated wood [51]. The NaCl, which created a micro-/nanoscale architecture into the coating, was removed one week later by rinsing with distilled water. The resulting wood surface was 100% natural, food safe and had a contact angle with water of 161°, but was not considered superhydrophobic as its roll-off angle was extremely high. Zhang et al. designed a biogel coating based on chitosan, gelatin and glycerol that had quite a low adhesion (1.4 MPa), but some very interesting features [52]. It could completely heal medium damages under heating, be reused after being scrapped from the wood

and dissolved in water, and be colored with water-soluble dyes. A 3-layers coating was produced on beech wood with a polydopamine primer, an hydroxyapatite second layer, and a chitosan topcoat made from shrimp wastes [53]. The composite coating showed good hydrophobicity (contact angle = 130°), photostability and resistance to seawater. After 6 months of immersion in the sea, the treated samples showed lower chemical degradation, color changes (E* = 12.68 vs. 22.24) and damages from barnacles than the controls samples. Liu and Hu prepared polystyrene colloidal microspheres with different acrylate-based copolymers [54]. Once casted on aspen, the very densely arranged microsphere exhibited different colors, such as green, orange and red. Other unique organic coatings were developed by using materials, such as chitosan oligomers, vegetable oil, castor oil and lignin to protect wood from decay fungi [55], fire [56], and photodegradation [57,58].

### 3.1.2. Additives in Organic Coatings

While it was shown in the previous section that organic wood coatings could be improved by using the appropriate reactives, another great way to obtain performant coatings is through the inclusion of functional additives. Organic, bio-based materials have been the subject of much research over the last 5 years. They are very interesting substances for wood protection, as they biodegrade upon leaching. Cellulose nanocrystals and nanofibrils received a lot of attention due to their potential to improve the mechanical properties of the softer oil- or resin-based organic coatings [59–63]. Tian et al. prepared a renewable UV-cured polyester methacrylate coating based on L-lactide and ε-caprolactone, containing 0% to 7.5% of cellulose nanocrystals (CNCs) [64]. The properties of the resulting composite coating changed proportionally to the concentration of CNCs, with an increase in bending strength, bending modulus, hardness and water contact angle, but a decreased tensile strength and elongation at break. At a 7.5% content of CNCs, the coating had a grade 3 adhesion, a 5H pencil hardness and a 103° contact angle. Veigel et al. incorporated 1% of cellulose nanofribrils (CNFs) to linseed oil after modification with acetic anhydride and (2-dodecen-1-yl)succinic anhydride to increase their solubility [65]. While the beech substrate coated with this varnish had the same initial hydrophobicity as the samples treated without the CNFs, the reduction of the hydrophobicity caused by multiple cycles of abrasion with a Taber Abraser was much slower for the CNFs containing formulations, showing a greatly reduced loss of oil. Kaboorani et al. modified cellulose nanocrystals (CNCs) with hexadecyltrimethylammonium bromide to improve their compatibility with a UV-cured acrylic resin [66–68]. They found that a loading of 3% of CNCs significantly enhanced the pencil hardness, tensile strength, modulus of elasticity, and thermal stability of the coating, while reducing its mass loss following abrasion and water vapor uptake and transmission rate. Cheng et al. found that adding CNCs and silver nanoparticles to a polyurethane coating exhibited a synergistic effect toward the antimicrobial properties of the coating, while also improving its adhesion [69]. Tree extracts were another type of additives that received a lot of attention, this time to imbue wood with better photostability [70,71]. Acrylic coatings containing condensed tannins and modified tannins were prepared by Grigsby to protect radiata pine [72] before exposition to natural and accelerated weathering. A loading of less than 0.5% of tannins was sufficient to extend the coating's life up to 20% more than commercially available hindered amine light stabilizers (HALS) and phenolic stabilizers could. Tomak et al. coated Scots pine with water-based acrylics containing tannins from different woods species in the presence [73] or absence [74] of metallic oxides. They found that after 1512 h of artificial weathering, their coatings could outperform the commercial reference coating in terms of color changes and chemical degradation. It appeared that the lower concentrations of extractives were more effective against UV degradation, and that the interactions between the different tannin and oxides types were completely random. Waterborne acrylic containing CNF and bark extractives was prepared by Huang et al., which yielded both photostability and better mechanical properties (hardness and abrasion resistance) [75]. Finally, Yan et al. prepared acrylic wood coatings containing delignified wheat-straw powder, either raw or after calcination [76]. The

resulting coatings showed good resistance to molds, especially when the wheat straw was calcined.

Some non-bio-based, organic wood coating additives were also studied over the last five years, sometimes under the form of microcapsules. Zhu et al. prepared urea-formaldehyde microcapsules loaded with thermochromic material to make color-shifting wood under thermal stimuli [77]. A waterborne varnish containing 20% of the microcapsules showed an important color change toward the red and yellow, according to the CIELAB analysis, when heated between 31 °C and 37 °C. The color change was then perfectly reversible between 34 °C and 26 °C. Other workers studied the development of heat sensitive wood, either with microcapsules [78] or not [79,80]. A self-healing acrylic coating was prepared by Yan and Peng by encapsulating resin in urea-formaldehyde microcapsules [81]. They found that a loading of 4% of microcapsules was enough to imbue the coating with good self-healing capacities without affecting its mechanical properties. Similarly, Queant et al. encapsulated organic UV absorbers in calcium carbonate microspheres in order to protect them from degradation [82]. A 2500 h accelerated weathering test showed that wood coated with a transparent waterborne latex would suffer less color changes when the UV absorbers were encapsulated. Other uses for organic additives in wood coatings during the last five years included the enhancement of their mechanical properties [83] and the reduction of the oxygen inhibition [84].

Nanoparticles of metal oxides and silica as wood coating additives allow the preparation of surfaces with a wide array of functionalities. An interesting method to achieve superhydrophobicity was to modify the nanoparticles with a low surface free energy chemical before their incorporation into the coating [85–88]. The low surface free energy of both the modified nanoparticles and the resin would imbue water repellency, and the nanoparticles brought an appropriate micro-/nanoscale architecture to the coating, joining together the two requirements to achieve superhydrophobicity. Sevda et al. experimented the addition of $SbO_3$ and $TiO_2$ to an intumescent paint [89]. They noted that a loading of 2% of nanoparticles increased of LOI and greatly decreased the weight loss and smoke generation in comparison to the paint alone. Guo et al. prepared bio-sourced silica particles through the calcination of rice husk [90]. After modification with a silane coupling agent (KH-570), a 2% loading of the silica in a waterborne acrylic coating improved the elongation at break (244.72% to 303.06%), tensile strength (32.509 MPa to 48.673 MPa), modulus of elasticity (3.010 MPa to 6.672 MPa), and pencil hardness (1H to 2H) of the resulting coating. Other workers explored the possibilities of these compounds to improve the resistance of wood to decay fungi [91,92], black-stain fungi [93], and photodegradation [94], as well as to improve its mechanical properties [95,96].

In addition to metallic oxides and silica, inorganic compounds of mineral origin were used as additives to improve the properties of organic wood coatings. Atienza et al. used oyster shell powder to make a thermally stable acrylic coating [97]. Because the shells are made of 95%–98% of incombustible calcium carbonate, the addition of 75% of oyster shell powder to the coating increased the time of burning of the wood samples from 18.00 min to 29.67 min. Zeolites were also considered as potential fire retardants in a melamine-urea-formaldehyde resin containing ammonium polyphosphate [98]. Due to their very porous nature, most of the zeolites studied showed an appreciable decrease in $CO_2$ production. The ignition time was also greatly delayed, from 138 s for the resin containing only ammonium polyphosphate to 279 s with 3A zeolites. Although different zeolites performed the best on the different aspects of fire protection, 3A zeolites were, overall, the most performant. Other compounds studied as fire retardants were graphene [99] and sepiolites [100], which could also be used to make hydrophobic coatings. Kolya and Kang coated various species of hardwood with polyvinyl acetate coatings containing modified graphene oxide [101]. The graphene oxide, which had been reduced with $NaBH_4$ in presence of urotropin and further functionalized with poly (diallyl dimethylammonium chloride), slightly increased the water contact angle of the coating on most wood species, with average angles of 91.5° and 92.7° on the radial and cross-sectional face, respectively, as compared to 72.5°

and 80.1° for the polyvinyl acetate coating alone. Similarly, Chen et al. functionalized sepiolite with polysiloxane and mixed them with an epoxy [102]. They found that the hydrophobicity of the coating increased rapidly with higher sepiolite:epoxy ratios, with highly superhydrophobic (water contact angle = 166° and roll-off angle = 5°) at 7:5. The wood surface also exhibited good self-cleaning properties and the ability to separate water and oil.

3.1.3. Organic-Inorganic Composite Coatings

An extremely large share of the research into wood coatings, over the last five years, focused on the organic-inorganic composite coatings. This section differs from the additives in organic coatings for a few reasons: 1- the organic and inorganic parts, in this section, are not always blended together, 2- the organic part is often not a resin, and 3- the inorganic part is frequently the main component of the coating. Also, organosilicons are an important element of these coatings.

A method frequently encountered to improve the hydrophobicity of the wood surface, and often reach superhydrophobicity, was to use nanoparticles to form a proper micro-/nanoscale architecture and thereafter reduce its surface free energy. For this matter, an organic coating could be used to reach the desired low surface free energy [103,104]. A superhydrophobic wood surface was prepared by Lozhechnikova et al. after applying positively charged ZnO nanoparticles and a negatively charged carnauba wax on Norway spruce through layer-by-layer deposition [105]. Not only did the coating reach a 155° water contact angle (WCA), but it also displayed higher UV stability and moisture buffering. Lu et al. pretreated rubberwood with IPBC, an organic biocide, before dipping it in polystyrene and $SiO_2$ solutions [106]. At higher $SiO_2$ concentrations (2%), the WCA was 155.6° and the antiseptic performances of the IPBC were preserved by the coating, which reduced its leaching. A superhydrophobic coating with a high thermal energy storage capacity was designed by Kong et al. through spraying with mesoporous polydivinylbenzene nanotubes, fluorine-containing $SiO_2$ nanoparticles and paraffin wax [107]. Upon exposition to excessive heat, the wax trapped in the nanotube would melt to store thermal energy and later release this energy through crystallization. This kind of wood surface improvement strategy was also used by other workers to imbue wood photostability [108], molds resistance [109], self-healing [110], thermal stability [111], and improved adhesion of UV-curing coatings [112].

The hydrophobization of the micro-/nanoscale architecture could also be achieved by replacing the resins with low surface free energy components [113–132]. As a general rule, these coatings showed superhydrophobicity (contact angle > 150°, roll-off angle < 10°), high resistance to mechanical wear (abrasion, cutting, etc.), and sometimes properties such as chemical resistance and self-healing. Wang et al. developed an interesting method to grant superhydrophobicity to Chinese fir, where they obtained the desired micro-/nanoscale architecture simply by sanding the wood surfaces with a 240-grit sandpaper [133]. The surface free energy of the produced micro/nanoscale architecture was subsequently reduced by deposition of a fluoroalkylsilane/silica composite suspension to obtain a superhydrophobic surface with good abrasion resistance and self-healing capabilities. Guo et al. created a Mg-Al-layered double-hydroxide coating to improve the fire safety of birch wood via thermal deposition followed by hydrophobization with trimethoxy(1H,1H,2H,2H)heptadecafluorodecyl)silane [134]. The limiting oxygen index (LOI) of the coated wood increased from 18.9% to 39.1%, and its total smoke generation and total heat release decreased by 58% and 40%, respectively. Wang et al. coated poplar wood by dipping in polydopamine for 24 h, electroless Cu deposition for 12 h, and dipping in octadecylamine for 24 h [135]. Although the process was quite tedious, the resulting coating was extremely durable, keeping its superhydrophobicity even after degradation by UV light, acids, bases, organic solvents (n-hexane, acetone, ethanol, and DMF), and boiling water. Huang et al. modified nanofibrillated cellulose [136] and lignin-coated cellulose nanocrystals [137,138] coated wood with 1H,1H,2H,2H-perfluorooctyltrichlorosilane through chemical vapor deposition. The resulting wood surface showed high sandpaper

abrasion and UV resistance, as well as superhydrophobic and self-cleaning behaviors. The naturally hydrophobic micro/nanoscale structure of canna leaves [139] and rose petals [140] was re-created by Yang et al. through nanoimprint lithography. They first created a PDMS template of the canna leaves and rose petals, which was use to make a perfect copy of the said structure with $SiO_2$ and polyvinyl butyral. The copy could then be peeled from the template and stuck to the wood surface to reach superhydrophobicity. A similar strategy was used by Chen et al. to create superhydrophobic and magnetic wood surfaces based on the structure of taro leaves with $F_3O_4$ and PDMS [141]. Gan et al. also prepared superhydrophobic wood surfaces with a ferromagnetic behavior by dipping poplar samples in a solution of hydrophobized $CoFe_2O_4$ nanoparticles, which had a contact angle of 158°, high resistance to sandpaper abrasion, and improved microwave absorption properties [142].

Some organic-inorganic composite coatings were prepared simply by adding organic and inorganic moieties together on the wood surface. Wang et al. used tannic acid-$Fe^{3+}$ complexes in combination with silver nanoparticles to create a superhydrophobic coating [143]. The developed coating was highly durable, keeping a contact angle higher than 150° after UV exposition and degradation by HCl, NaOH, n-hexane, acetone, ethanol, DMF and boiling water. A magnetic wood coating based on chitosan, sodium phytate and nano-$Fe_3O_4$ was prepared by Tang and Fu through layer-by-layer deposition [144]. They found that paramagnetic wood with narrow and long magnetic hysteresis loops could be created with this method, the magnetic properties of the treated wood being directly related to the number of layers in the coating. Uddin et al. prepared a paste with $Mg(OH)_2$ and casein to improve Scots pine's resistance to fire [145]. The prepared wood surface had a delayed time to ignition (12.1 s to 30.4 s), a lower peak heat release (216 $kW/m^2$ to 119 $kW/m^2$) and a lower total heat release (79.5 $MJ/m^2$ to 53.3 $MJ/m^2$), as well as decreased smoke production and mass loss. Another fire-retardant coating was prepared by Xie et al. through dipping in different solutions containing graphene oxide and functional cellulose [146]. These coatings scored a V-0 rating in a vertical burning test, FH-1 or FH-2 rating in a horizontal burning test, had a greatly increased LOI, and could self-extinguish when removed from the flame source. In presence of moisture, they could even self-heal incisions with widths up to 320 μm.

### 3.1.4. Inorganic Coatings

Metallic oxides and silica played an important role in the last five years of research in the domain of wood coatings [147–149]. They were layered on wood surfaces with a variety of methods, including sol-gels [150–152]. Sol-gels with $SiO_2$, $TiO_2$, and $Fe^{3+}$ [153] or $Zr^{4+}$ [154] were used to make photostable wood surfaces with photocatalytic activity, granting them self-cleaning capabilities through the photodecomposition of organic pollutants. Qian et al. also used sol-gels to develop a coating based on microcapsules with a $Fe_3O_4/SiO_2$ shell and a phase changing material core to imbue energy storage and magnetism to poplar wood [155]. The hydrothermal growth and deposition of metallic oxides were also the subject of many publications [156–159]. Sun and Song casted $WO_3$ on poplar wood through hydrothermal in situ synthesis [160] or nanosheet deposition [161] to build photochromic wood. The resulting wood surfaces could reversibly change color after exposition to UV radiations, had better photostability and could be hydrophobized with 1H,1H,2H,2H-heptadecafluorodecyl)silane. Similarly, $MoO_3$ was hydrothermally grown or deposited on birch to yield photo-responsive wood with a blue shift when exposed to UV light [162,163]. Wang et al. prepared magnetic wood with fire-retardancy through the hydrothermal deposition of $MnFe_2O_4$ [164]. The initial burning time of the coated wood was delayed from 6 s to 20 s, its electromagnetic waves absorption capacity was improved, and it could additionally be hydrophobized with fluoroalkylsilanes [165]. A very interesting silica coating was prepared by Belykh et al. by mixing sodium liquid glass and black shale, which are byproducts from the fabrication of ferrosilicon and gold mining activities, respectively [166]. They found that good adhesion could be achieved by using 20%–35% of black shale, while 10%–25% yielded a good reduction of the mass loss

when exposed to fire. They also found that adding 1% of a synthetic foaming agent (PO-6) improved both the adhesion and fire resistance of the coating.

Among the other approaches to coat wood with inorganic materials, Pan et al. used an electroless plating method to coat poplar disks with nickel (Ni) [167]. They found that the disks' resistance would decrease from 12 $\Omega$ to 1 $\Omega$ within the first 5 minutes of electroless plating, that the hydrophobicity of the coated wood slightly increased, and that an electromagnetic shielding effectiveness between 55 Db and 65 Db could be achieved. Similar methods were also studied to coat poplar with Cu-Ni [168] and Ni-P [169] composites. A superhydrophobic wood surface was created by Wang et al. by deposing copper on a wood substrate with a vacuum evaporator, followed by the growth of a silver layer through immersion in a $AgNO_3$ solution [170]. The prepared wood surface, which had a contact angle of 160.5° and a roll-off angle near 0°, could keep its superhydrophobicity after 100 cycles of tape abrasion or 200 cm of sandpaper abrasion with a 50 g weight. Hydrophobic wood surfaces were prepared by Łukawski et al. via drop casting and dipping with different solutions of carbon black, graphene, and carbon nanotubes [171]. Concentrations as low as 0.05 $g/m^2$ of nanomaterials were sufficient to reach very high water contact angles (up to 143°), although superhydrophobicity was not achieved. They also found that, while carbon nanomaterials do not make covalent bounds with wood, the coatings' resistance to sandpaper was quite good. Yuan et al. used hydrothermal deposition of graphitic carbon nitride nanosheets to improve the photostability of poplar wood [172]. They found that the nanosheets could absorb 90% of the UVA and UVB, substantially reducing the color changes after accelerated weathering. Furthermore, TG and DTG showed an improvement of the thermal stability of the coated wood. Another use of inorganic nanosheets as wood coatings was explored by Liu et al. who coated cedar wood with boron nitride nanosheet to improve its performance against fire [173]. The obtained wood surface showed a good thermal stability after 60 s of exposition to a lighter and an improved resistance to oxidation.

### 3.2. Wood Surface Modification

Instead of improving wood surfaces through the addition of an outer layer, different modification methods allow for the enhancement of the wood's properties directly. These methods involve the chemical modification of the wood surfaces by different means to improve their wettability, decay-fungi resistance, photostability, and more.

### 3.2.1. Plasma Modification

An environmentally-friendly way to modify the surface of wood is through plasma treatments. Plasmas are highly reactive chemical environments with interesting features such as scalability and the absence of solvents [174]. They exist under various forms, which can be thermal or non-thermal [175]. In the case of wood protection, however, non-thermal plasmas are preferred to avoid its thermal degradation. An interesting use of plasmas in wood surface modification is to oxidize its polymeric constituents to increase its surface free energy and wettability. Over the last 5 years, many workers explored this application to increase the interactions between wood and adhesives or coatings [176–187]. They found that using reactive carrier gases such as air and $O_2$ would lead to the creation of polar groups such as carboxyl and carbonyl at the surface of the treated wood, increasing its hydrophilicity and surface free energy. The plasma-treated wood would therefore have much lower contact angles with water and water-based coating solutions, showing an enhanced wettability, as well as a faster and deeper absorption of those liquids. As a result, the adhesion of coatings and adhesives would often be greater on treated wood than on its untreated counterpart.

Žigon et al. [188] and Žigonand Dahle [189] used a floating electrode dielectric barrier discharge (FE-DBD) plasma to treat Norway spruce following a short dip in NaCl solutions. They found that the electric conductivity and the intensity of the discharge were both increased by the NaCl, which led to an enhanced wetting of the wood following the plasma treatment. As a result, the contact angle with water and a water-based coating were lower

and the tensile strength of the coating was improved. In another study, Žigon et al. used different adhesives to bind beech to aluminum and steel after treating both the wood and metals with a FE-DBD plasma [180]. The surface free energy of all the substrates increased following the plasma treatment, which led to higher tensile shear strength in most of the studied scenarios. After noting the poor adhesion of a water-based primer to beech veneers and of a water-based topcoat to an oily UV-cured primer, Peng and Zhang treated both the wood veneers and the UV-cured primer with a DBD plasma before laying the subsequent coat [190]. They found that the wettability of the beech veneers and the primer increased following the plasma treatment, which led to a large enhancement of their adhesion. Similarly, Dahle et al. dip-coated pine samples with polystyrene microspheres before proceeding to plasma modification of the coating [191]. As hydroxyl and carbonyl groups were created on the polystyrene layer, the wood surface became superhydrophilic and may eventually be used as a primer for further plasma polymerization. In an effort to improve the fireproofing of wood, Gospodinova and Dineff studied the effect of a plasma treatment on the absorption of fire-retardant solutions [192]. They found a substantial increase in the surface free energy and of the penetration-spreading parameter after the plasma treatment, but also a rapid decrease in these two variables during a post-treatment storage. Finally, Volokitin et al. used a thermal plasma treatment to mimic a thermal treatment on the surface of pine and birch [193]. Similar to typical thermal treatments, the modified samples were darker, their water contact angle increased, and their water absorption declined.

Other uses for plasma treatments in wood-surface protection are the grafting of chemical components and the creation of thin coatings. Due to their highly energetic and reactive nature, plasmas can melt metallic particles or break down organic molecules into reactive moieties while creating radicals on the surface of wood, which allows the growth of thin coating layers or the functionalization of wood surfaces [194,195]. Thereby, a Zn/ZnO thin coating was deposited on beech wood by Wallenhorst et al. through the cold plasma spraying of Zn microparticles with air as the process gas [195]. After 50 h of exposition to UV radiations, the thickest coating almost completely inhibited to color changes (E* ≈ 0), while uncoated wood had a E* of 10. They also found that the coating could protect a polyurethane topcoat from photodegradation. Similarly, Profili et al. prepared a hydrophobic $ZnO/SiO_2$ composite coating on sugar maple with a DBD plasma [196]. The coating, formed by the embedding of ZnO particles in a $SiO_2$ layer, displayed a static water contact angle of 100°, while the untreated samples quickly absorbed the water droplets. A superamphiphobic coating was casted on birch wood by Tuominen et al. by depositing titanium nanoparticles with a liquid flame spray followed by the plasma polymerization of perfluorohexane [197]. The coating, which displayed contact angles > 160° with water, ethylene glycol, diiodomethane, and olive oil, was also highly resistant as its wetting properties were still intact after a 500,000 water drops impact test. Furthermore, good self-cleaning properties were noted with both water and oil.

The hydrophobization of wood surfaces through plasma treatments with fluoroalkanes and organosilicons also received some attention over the last five years. Notably, de Cademartori et al. created a fluorocarbon film on white spruce and Brazilian cedar by polymerizing octofluoropropane with a DBD plasma [198]. They found that longer treatment periods would lead to higher hydrophobicity, with optimal water contact angles of 135.2° and 129.8° on the spruce and cedar, respectively. Levasseur et al. also used $C_3F_8$ to improve the hydrophobicity of sugar maple wood by DBD plasma with inert gases (Ar and $N_2$) [199]. They noted that the hydrophobicity of the obtained surfaces was directly linked to the voltage of the plasma, which could yield a 140° static water contact angle at the highest voltage (10 kV). After letting the coating age for 125 days under uncontrolled conditions, the wetting properties of the wood surface remained unchanged. The possibility to increase or decrease the wettability of wood through cold remote ($N_2 +O_2$) plasma was studied by Bigan and Mutel [200]. They noted that the water absorption of different plasma-treated wood species could significantly increase (up to 5.5 timesin the case of

beech) by using the plasma treatment alone, but that adding 1,1,3,3-tetramethyldisiloxane could make the wood superhydrophobic. Wood coatings with both hydrophobicity and good thermal stability were prepared by Sohbatzadeh et al. [201] and Chen et al. [194,202] through the plasma polymerization of hexamethyldisiloxane. Wood surfaces with lower surface free energy and increased roughness were obtained, yielding water contact angles as high as 138°. Many other studies were conducted over the last five years to improve the properties of wood surface through plasma treatments, including polyester powder [203], polyester with aluminium coated silver and bismuth oxide [204] or $TiO_2$ [205], copper and aluminium microparticles with an acrylic binder [206], ZnO [207], $TiO_2$ [208], and various biocidal precursors [209].

### 3.2.2. Other Surface Modification Methods

Beside plasma treatments, many methods were used, over the last years, to improve the properties of wood surfaces through modification. Herein, those methods are classified into two categories: chemical methods and carbonization methods. The chemical methods relied on chemical reactions or interactions to tone the properties of the outmost surface of the treated wood; a slight penetration of the chemicals into the wood was considered a surface impregnation, which will be reviewed in the next section. The chemical grafting of chemicals on the surface of wood involves the creation of a covalent bond between the wood cell wall and the modifying agent. This method allows to improve the surface properties of wood while reducing the leaching of the chemicals [210]. Wang et al. grafted poly(2- (perfluorooctyl)ethyl methacrylate) on the surface of Chinese fir by atom transfer radical polymerization [211]. The modified wood had a strong superhydrophobic behavior, which was only slightly affected by finger-wiping and tape-adhesion abrasion tests. Furthermore, the treated wood showed excellent resistance to the mold *Aspergillus niger* and self-cleaning properties. A similar method was used by Sharma et al. to graft acetonitrile and ethyl acrylate on pine wood [212]. Under optimal conditions, they obtained a percentage grafting of 85.34%, which greatly reduced the swelling of the treated wood in different solvents and solutions, as well as its weight loss when dipped in strong bases and acids. An environmentally-friendly treated was developed by Filgueira et al. as they grafted modified *Pinus radiata* tannins and condensed tannins on beech wood through the action of a laccase enzyme [213]. They found that treating wood this way, under an alkaline medium (pH = 10), would reduce its water absorption over 72 h by 20% and reduce the leaching of the treatment by 76%. Song et al. modified the surface of balsa wood by dipping samples for 60 s in aqueous solutions containing 0.75% of different salts [214]. They found that the metal ions, particularly $Zr^{4+}$, could attain a water contact angle up to 145° through the creation of a microstructure and crosslinking. However, the durability of such treatment seems rather low, as the contact angle dropped to 138° after 14 days ambient conditions. The fluorination of silver fir and Douglas fir with gaseous $F_2$ was studied by Pouzet et al. [215,216]. They found that the treatment would substitute hydroxyl groups from the cell wall polymers for fluorine, reducing the surface free energy of the treated wood. As a result, the water contact angle increased up to 120°, the water absorption decreased drastically, and the treatment was still as hydrophobic two years later. However, while short treatment times had only a low effect on the integrity and color of the treated wood, treatments of 20 min led to a severe degradation of the tracheids and browning of the wood surfaces. In another study, they also noted that a torrefaction post-treatment would help to purge the HF produced by the reaction of $F_2$ with wood and further slow down the ingress of water without causing any defluorination of the treated wood [217]. The combination of a laccase enzyme surface treatment and pressure impregnation of copper(II) sulfate pentahydrate was explored by Gabrič et al. [218]. They found that a laccase pre-treatment would make the cell walls of the wood swell, preventing its impregnation; however, using the laccase as a post-treatment would greatly reduce the leaching of the copper. Furthermore, the laccase treatment alone could reduce the mass loss due to the degradation by brown- and white-rot fungi by roughly a third. More chemicals were studied over the course of the last

five years to modify the surface of wood, including aminoborates [219], methanol [220], cellulase [221,222], poly(methylhydrogen)siloxane [223], and chitosan [224].

As the name implies, the carbonization methods revolve around charring the surface of wood to modify its properties. The use of $CO_2$ lasers with radiation doses up to 75 J/cm$^2$ was studied in several publications, as well as its effect on the tensile strength of adhesives, the resistance to molds of the treated wood and its surface free energy [225–227]. The authors found that high radiation doses would increase the wood surface blackening, as a result of the carbonization, as well as its resistance to molds. However, they noted that the treatment was only effective against *Aspergillus niger*. Studies of the surface properties showed that the loss of hydroxyl groups reduced the surface free energy of the treated wood, which decreased the tensile strength of polyurethane and polyvinyl acetate adhesives. Other authors explored the carbonization of wood by pressing a single surface of the treated wood with a hot metal plate at different temperatures (220 °C to 400 °C) for different periods (30 s to 2 h) [228–230]. They observed highly modified moisture-related behaviors, with higher water contact angles, lower water absorption, and lower equilibrium moisture contents (EMC). As a result of the reduced EMC, the modulus of rupture of the charred wood was also slightly higher. A similar study was conducted by Akçay et al., wherein, pine and beech surfaces were carbonized with a blow torch to improve the wood's resistance to white- and brown-rot fungi [231].

*3.3. Wood Surface Impregnation*

The properties of the wood surfaces can be improved with a shallow impregnation of chemicals. It can be achieved by different means, from brushing and very short dippings (few seconds) to longer dippings (few hours) and single face vacuum impregnation. Very few publications on the subject of impregnation presented details about the impregnation depth of the chemicals or their distribution into the wood; consequently, different parameters were taken into consideration to decide if a method would be considered as a surface impregnation: the use or not of pressure/vacuum, the duration of the treatment, the size of the samples, whether the samples were completely or partially covered by the treatment, and the weight gain. Petrič described surface impregnation as the impregnation of the first few millimeters of the cross-section of wood [175]. While this definition was used as the basis to classify the treatments as surface impregnation, it seemed rather ambiguous, as some hard-to-treat species can only be treated by a few millimeters in the cross-section, even with a vacuum/pressure process. Accordingly, treatments that would allow high longitudinal penetration were also rejected.

A primary way to treat wood by surface impregnation was through the insertion reactive material. Triquet et al. chemically increased the surface density of various hardwood species by in situ polymerization of acrylate monomers [232]. The monomers were vacuum impregnated for 150 s after being dropped on a single surface of wood, which was followed by electron-beam polymerization. The density of the treated wood increased by nearly 200 kg/m$^3$ near the surface, which lead to an augmentation of the Brinell hardness. Different workers studied the possibility of reducing the set-recovery of unilaterally compressed wood with the impregnation of chemical agents. While the compression of wood itself is not a subject of this review, the effect of a surface pre-treatment on its durability was deemed appropriate. Wu et al. impregnated poplar wood with a reactive waterborne acrylic resin by immerging a quarter of the wood blocks into the resin solution and applying a vacuum, leading to weight percent gains ranging from 1.1% to 4.7% [233]. Afterward, the impregnated surface was densified with a hot press under different temperatures ranging from 150 °C to 180 °C. At the highest loading of resin, the set-recovery of the impregnated wood was only 1.8%, while the control samples reached 73.0%. Similarly, Han partially soaked Scots pine for different durations in a furfuryl alcohol solution containing a maleic anhydride catalyst before pressing a single surface with a hot metal plate [234]. Under optimized conditions, the set-recovery of the densified wood decreased from 60% to 14%. Various impregnation agents were studied by Neyses et al. to achieve the same goal,

although the results were not as satisfying [235]. Lafond et al. improved the embedment capacity of black spruce connectors through the impregnation of acrylates [236]. They found that a chemical retention of 7% could improve the bearing strength of the connector by 48% and their stiffness by 27%. Finally, the development of colored wood surfaces through the creation of complexes between phenolic extractives and metal ions was explored by Dagher et al. [237]. After simply applying a 1% ferric sulfate solution on the surface of different hardwoods with a foam roller applicator, different colors were developed for each species according to their phenolic extractives content.

Another way to protect wood by surface impregnation is to simply insert protective agents slightly under its surface. Harandi et al. brushed 5% and 10% solutions of poly(vinyl butyral-co-vinyl alcohol-co-vinyl acetate) (PVBVA) on silver fir to improve its mechanical properties [238]. They found that both solutions increased the water contact angle to 90%, reaching hydrophobicity, and improved the modulus of rupture (MOR), modulus of elasticity (MOE), plastic hardness and Martens hardness of the treated specimens. Although the more viscous 10% solution took more time to be absorb, it yielded equivalent MOR and MOE, as well as higher plastic hardness and Martens hardness than the 5% solution. Kumar et al. dipped Norway spruce blocks in a 1% solution of octadecyltrichlorosilane for 30 min to 120 min, yielding from 0.7% to 2.4% weight percentage gains [239,240]. They found that the treated specimens had very high static water contact angles (140–150°), a negligible water absorption through immersion, a lowered equilibrium moisture content when exposed to a high relative humidity (95%), an increased dimensional stability, and a reduced mass loss when exposed to the brown-rot fungi *Coniophora puteana*. The impregnation of poplar wood with $K_2CO_3$ and $SiO_2$ solutions to improve its fire-retardancy was studied by He et al. [241]. They found that the limiting oxygen index (LOI) increased from 20.5% to 33.5% prior to the treatment, and only decreased to 30.5% after leaching. Thermogravimetric measurements showed that the mass loss decreased during the charring phase (63.2% to 47.4%) and the calcining stage (34.8% to 24.3%), while the char generation increased. Thermochromic wood veneers were prepared by Zhu et al. through the ultrasonic impregnation of a thermochromic dye and a color developer [242]. The treated wood, which was very dark, could return to its original color between 28 °C and 38 °C, while the discoloration was reversible between 34 °C and 22 °C. Other publications reported the surface impregnation of chemicals to decrease the wettability [57], flammability [243], mold [244,245] and mildew [246] degradation, and dimensional instability [247] of various wood substrates.

## 4. Discussion

As previously presented, a very wide variety of methods can be used to protect the surface of wood, combining various techniques and chemicals. However, in order to have a concrete beneficial effect on our lives, new wood protection innovations must be used in real buildings. The use of wood in construction is however a fairly specific domain, where the application determines which properties (photostability, hydrophobicity, fire-retardancy, self-cleaning etc.) are needed and which sources of wear and degradation can be encountered. Different considerations must be taken into account, such as governmental regulations or the treatment's cost, duration, durability, and ecologic footprint. This section of the review will be dedicated to discussing some of these specificities to identify which trends and treatments could be adequate in different situations.

### 4.1. Normative Aspects

A first prerequisite for any technology to be applied in real-life applications is to satisfy different local norms and regulations. As an aegis of the public's safety, these regulations emphasize the need for wood-based products to be performant and legitimize the development of new and innovative treatments. Different aspects of wood protection are supervised by authorities, such as fire safety and decay.

Fire hazards present a huge stake in lumber construction, particularly for high-reaching buildings[26]. They often compel construction companies to hide the wood behind noncombustible material, restricting the use of apparent timber. Specific guidelines dictate the way wood buildings must be constructed and important restrictions are applied to ensure the safety of the construction workers and the residents [248]. They include criteria, such as the charring rate and the preservation of the load-bearing capacity of the building in the event of a fire. In order to reach these targets, wood surface modifications can become key elements to create high quality materials. Satisfyingly, many workers over the last five years tackled the challenge of improving the fire safety of wood through surface improvements. In fact, almost every trend presented in this review contained at least one treatment aimed at reducing the hazards of flames.

Organic coatings were particularly studied for this subject, either by the inclusion of phosphorous reactives [30–35] or additives [63,85,89,97–100]. Different mechanisms were tested to promote the fire safety of wood such as an increased char formation, a delayed time to ignition, an increased limiting of the oxygen index (LOI), and a lower smoke release. Phosphorous reactives were usually quite effective at increasing the char formation and the LOI. The formation of char at the surface of wood is an excellent mechanism to improve its fire safety, as it creates an insulating layer that protects its inner section and decelerates the loss of mechanical properties [249]. This kind of protection is actually very interesting as a category of fire-retardant coatings; the intumescent coatings were designed to utilize this mechanism to protect wood from flame sources by transforming into a porous insulating char barrier upon heating [250]. The LOI is also an important aspect of fire protection; as an indicator of the flammability of a material, it indicates how readily the substrate will ignite and extinguish [251]. A LOI as high as 33% was obtained by Lokhande et al., while removing the phosphorous moieties from the coating would reduce this value to 23% [34]. As a whole, it shows that phosphorous reactives offer great potential for producing fireproof coatings.

Organic coating additives were very efficient at reducing the aspects related to the combustion of the wooden substrate such as the ignition time, the burning time, and the heat release. Although many different chemicals were used as additives to increase the thermal stability of the coatings, they presented similar properties as they had a mineral nature in common. As such, they displayed low flammability, reactivity with oxygen, and thermal conductivity, resulting in a less intense combustion. An interesting coating was developed by Wu et al. through the addition of zeolites in a melamine-urea-formaldehyde resin containing ammonium polyphosphate [98]. As very porous material with high surface area, zeolites can adsorb high quantities of gases; by using the right zeolites, the release rate of CO and $CO_2$ could be decreased.

Other approaches were investigated to improve the thermal stability of wood surfaces. Among those, inorganic coatings were extensively studied, particularly through hydrothermal deposition [122,164,172,173]. Their impact on the combustibility of wood was similar to that of organic coating additives, as their chemical nature and properties are similar. Accordingly, higher LOI, delayed ignition and lower mass loss could be achieved with these simple treatments. High thermal stability was also achieved with surface impregnation of chemicals [241,243], which is perfectly coherent since pressure impregnation of fire-retardant compounds is conventional in wood protection. However, many issues can be encountered with fire-retardant pressure impregnated wood such as high chemical uptake and leaching, moisture sensitivity, and reduced mechanical strength [250,252]. Thereby, the methods focusing on the surface impregnation did not only simplify the treatment procedures, but they also showed good performances at a lower product retention. Moreover, the combination of silica and $K_2CO_3$ presented by He et al. displayed a fairly lower leaching when compared to the $K_2CO_3$ alone [241].

Another crucial aspect of wood protection is the use category. The need for protection of timber is directly related to the source of degradation it is exposed to; accordingly, regulations exist to ensure that wood exposed to certain elements, particularly wood eating

organisms, will be able to withstand the degradation. An example of such classification was prepared by the American Wood Protection Association (AWPA) with uses categories such as "interior dry" (UC1), "exterior above ground, coated with rapid water runoff" (UC3A), "ground contact, general use" (UC4A), and "Marine Use, Northern Waters (Salt or Brackish Water)" (UC5A) [253]. While using natural resistant species can help to attain satisfying levels of resistance, using proper treatments allows for reaching higher-use categories with cheaper and abundant species.

As a biomaterial mostly composed of carbohydrates, wood can be consumed by different living organisms such as molds, decay fungi, insects, and marine borers, which leads to various complications [5]. While molds feed on small molecules contained in the parenchyma such as fatty acids and starches, leaving the polymeric constituents intact, they create unsightly discolorations on the surface of the wood [4]. While this aspect does not represent a threat to human health, it does reduce the lifespan of timber, which is of ecological and economical importance. Moreover, they also release spores into the air, which represents an important health hazard. Decay fungi, on the other hand, do feed on the structural polymers of wood, creating not only a discoloration of the surface of wood, but also mechanical damage [254]. As a result, the chemical properties of the timber are modified, and its mechanical strength is greatly impaired. In order for molds and decay fungi to grow, different conditions must be met. The most important one is having a sufficient amount of moisture in the wood, which can be as low 12.3% of the dry mass of wood depending on the fungus [255]. Additionally, moisture changes in wood are linked to dimensional variations, which are directly responsible for mechanical defects like cupping and cracking [256]. While wood used indoor is rarely subjected to such a high moisture content, it is a different reality when it comes to wood exposed outdoor.

The significance of this issue can easily be visualized by the huge number of publications targeted at increasing the hydrophobicity of the wood surfaces over the last five years. Because the exposure to rain represents such an abundant source of water for wood to absorb, a lot of treatments were prepared to imbue wood surfaces with (super)hydrophobicity. The most prominent trend surrounding the hydrophobization of wood was through the deposition of inorganic nanoparticles, either before or after modification with low surface free energy compounds. In order to achieve superhydrophobicity, both a high water contact angle (>150°) and a low roll-off angle (<10°) are required [20]. An effective method to obtain very high contact angles is to create a surface with variations at a microscopic scale covered with nanosized indentations (micro-/nanoscale architecture) [257]. This structure can easily be achieved with nanoparticles, although a very innovative approach was developed by Wang et al. to prepare this architecture simply by sanding the wood substrate with a 240-grit sandpaper [133]. For water to easily roll over the surface, its surface free energy should be as low as possible to minimize its interactions with water. The most frequent compounds studied to decrease the surface energy of the wood surfaces were fluorinated moieties and organosilicons, but could also include organic resins, waxes, or organic acids [88,131].

The large quantity of superhydrophobic coatings developed over the last 5 years showed the importance to protect wood from water, but also that it is a well-understood subject. Consequently, attention should be given to coatings that could combine superhydrophobicity with other important properties. As a result of their precise micro-/nanoscale architectures, superhydrophobic coatings tend to be fragile and lose their superhydrophobicity once subjected to mechanical wear [258]. Many workers studied the effect of sandpaper abrasion on the hydrophobicity of their coatings, which almost always resulted in a consequent decrease in the water contact angle. However, some coatings did perform quite well when exposed to sandpaper abrasion [118,123,136–138], showing that superhydrophobicity can be combined with durability. In addition to high-wear resistance, the longevity of different superhydrophobic coatings were further improved through self-healing [110,133], which allows to recover the micro-/nanoscale architecture following mild physical damage. Other elements of the protection of wood were also incorporated to

superhydrophobic coatings including photostability [105,205,207,208], resistance to chemicals [88], and thermal stability [85,111]. Beside aspects related to the preservation of timber, some publications described superhydrophobic coatings with interesting new functionalities. Kong et al. spray coated a solution containing a paraffin wax, polydivinylbenzene nanotubes, and fluorine-containing $SiO_2$ nanoparticules on a wood substrate [107]. In addition to its superhydrophobicity, the wax absorbed in the nanotubes could act as an energy storage when exposed to high temperatures, absorbing excessive heat through fusion; the heat could later be released as the wax would crystalize. Poplar wood with a ferromagnetic behavior was prepared by Gan et al. after applying a layer of epoxy primer and dipping it in a solution of $CoFe_2O_4$ nanoparticles hydrophobized with 1H,1H,2H,2H-perfluorodecyltriethoxysilane [142]. Beside a high water contact angle and resistance to sandpaper abrasion, the prepared wood surface had significantly improved microwaves absorption properties and a minimum reflection loss of $-12.3$ dB.

While rain represents an important aspect of the protection of wood against water, it is possible for wood to reach a sufficient moisture content to promote the growth of molds and decay fungi simply from the air moisture. Consequently, even wood sheltered from the rain may reach a critical water content without proper protection. Sadly, very few treatments presented in this review could decrease the absorption of moisture, or at least very few were tested for this application. Qu et al. dip-coated Chinese fir in different sol-gels made from tetraethoxysilane (TEOS) and methyltriethoxysilane (MTES) [150]. After 30d of conditioning at 90% relative humidity and 30 °C, the resulting coatings absorbed much less moisture than their uncoated counterparts, with mass gains reduced from more than 10% to 1%. Another coating was developed by Lozhechnikova et al. through the layer-by-layer application of carnauba wax and ZnO nanoparticles on Norway spruce [105]. They found that a single bilayer would suffice to increase the moisture buffer value from 1.12 (untreated spruce) to 1.46, without any gain when adding more layers. Pouzet et al. decreased the equilibrium moisture content of silver fir and Douglas fir by fluorinating their surface with $F_2$ [216]. After exposing the treated wood to either 30% or 60% RH at 30 °C, they found that the EMC could be up to 20% lower than the untreated specimens. Finally, the surface impregnation of octadecyltrichlorosilane (OTS) in Norway spruce was studied by Kumar et al. to improve both its hydrophobicity and resistance to brown rots [239,240]. They found that the OTS could decrease both the absorption rate of moisture and the equilibrium moisture content of the treated wood, while decreasing the mass loss from the brown-rot fungi *Coniophora puteana* from 48% to 15%. As a crucial aspect of the hydrophobization of wood exposed outdoor, the authors believe that more studies should consider including the sorption of moisture in the future.

Of course, when wood cannot be efficiently protected from molds and fungal growth, another option is to treat it directly with biocidal material. Many of the trends presented in this review applied such treatments, including organic coatings [92], organic coating additives, surface carbonization [225,226], and chemical surface modification. Organic coatings containing bio-based materials were presented by many authors, including an antiseptic UV-cured coating made of citric acid and glycidyl methacrylate [43] and a composite coating based on a castor oil maleic anhydride adduct, epoxidized vegetable oil et 5−Bromosalicylic acid [55]. Lazim et al. presented noteworthy coatings prepared with *Dioscorea hispida sp.* starch and polyvinyl alcohol [41] or polyacrylamide [42] to reduce the degradation of timber exposed to the white-rot fungi *G. trabeum* and *C. versicolor*. Interestingly, the coatings could reduce the mass loss after 120 days of exposition to the fungi by more than 75% despite being based on carbohydrates themselves. Because of the harsh conditions and marine borers, the protection of wood in marine environments (Class 5) is extremely difficult and usually relies on the pressure impregnation of toxic chemicals or whole wood modification, or wood is usually destroyed within a year [259,260]. Nonetheless, Esfandiar et al. were able to develop a coating that could resist satisfyingly in sea water, reduce the photodegradation of the underlying timber, and reduce the degradation of wood by the barnacles [53]. While their method was extremely tedious (24 h immersion

in polydopamine, followed by 14d in hydroxyapatite, and 4 h in chitosan), it opens the way for a new generation of seawater-resistant coatings.

Among the additives studied to prevent biodegradation, Ag nanoparticles proved to be efficient against black-stain fungi [93] and bacteria [69]. In their study on the potency of Ag nanoparticles in acrylic latexes against different black-stain fungi, Boivin et al. noted that a concentration as low as 0.03% was sufficient to completely inhibit the staining by *S. pityophila* and *E. nigrum*, while higher concentration would compromise their dispersion in the film, showing how little of this ingredient could potentially be sufficient. Conversely, Cheng et al. found that much higher doses of Ag nanoparticles were needed in waterborne polyurethane coatings to provide a satisfying antimicrobial activity, although a good synergy was found with nanocrystalline cellulose [69]. As concerns exist around the use of metallic nanoparticles, a more environmentally-friendly method was proposed by Yan et al. to use wheat-straw powder et calcined wheat-straw powder after lignin removal as an antimould agent [76].

A major problem with wood protection is the loss of the chemicals, either through mechanical damages or leaching. Even when wood is satisfyingly protected against biodegradation following the treatment, exposure to the elements can deplete the treatment and make it useless after a while. Some good technologies were developed in the last few years to reduce this problem such as the surface modification of wood, and more precisely the chemical grafting. Wang et al. chemically grafted poly(2-(perfluorooctyl)ethyl methacrylate on Chinese fir by atom transfer radical polymerization [211]. As a result, the treated wood displayed superhydrophobicity, self-cleaning, resistance to *A. niger*, and extremely high durability to different forms of abrasion. A similar strategy was employed by Sharma et al. to graft acetonitrile and ethyl acrylate on pine samples to improve their antiseptic properties [212]. Additionally, the chemical grafting of hydroxypropylated *Pinus radiata* bark tannins with laccase enzyme allowed Filgueira et al. to reduce the leaching of the tannins by more than 40% in acid, neutral and alkaline mediums [213].

Other methods were studied over the last five years to limit the loss of protecting agents. Lu et al. showed that using a coating could contribute to the reduction of the leaching of impregnated material [106]. Indeed, by dipping rubberwood pretreated with 3-iodo-2-propyl-butyl carbamate (IPBC) in polystyrene and silica, they found that the efficiency of IPBC would increase as it cannot leach from the samples and allow the penetration of molds. Pantano et al. added copper-amine, a typical wood preservation agent, as an additive in an acrylic paint [91]. While they found that the leaching of copper-amine in the coating was almost 100-fold lower than the impregnated samples, decay tests, however, indicated that it could not properly protect wood from decay fungi. Nonetheless, coatings are not always needed to reduce the leaching of impregnated material, as Gabric et al. found that treating wood with a laccase enzyme could induce the swelling of the cell walls, thereby preventing the chemicals from escaping [218].

As presented in this section, the major risks associated with wood are related to its inherent combustibility and biodegradability. Therefore, from a normative point of view, an ideal treatment could protect wood from both fire and biodegradation. However, none of the treatments presented in this review could achieve both directly. However, some of them could improve the fire safety as well as greatly reduce the absorption of water, which can help to prevent the growth on molds and decay fungi [85,111]. Guo et al. prepared a Mg-Al-layered double-hydroxide inorganic coating on birch wood through a hydrothermal process [134]. The resulting wood surface had a greatly increased LOI (from 18.9% to 39.1%), as well as lessened total smoke (58%) and heat (40%) releases. Thereafter, the coating could be modified with trimethoxy(1H,1H,2H,2H-heptadecafluorodecyl)silane to reach superhydrophobicity. While the water contact angle on the Mg-Al coated wood was around 50° and decreasing over time, it was stable at more than 150° after modification, showing low water absorption. However, the study did not mention if the coating could efficiently protect wood from air moisture. Similarly, Kong et al. developed a superhydrophobic coating with high thermal stability by growing ZnO nanorods on Chinese fir through an

hydrothermal process followed by hydrophobization with stearic acid [122]. Furthermore, this coating also displayed a higher photostability, which is crucial for clear coatings used outdoor. Finally, Sohbatzadeh et al. used an atmospheric pressure dielectric barrier discharge plasma to create a rougher surface on fir wood, while depositing a PDMS-like coating with hexamethyldisiloxane as a precursor [201]. The resulting coating increased the water contact angle of the superhydrophilic fir from 0° to 138°, while retarding the apparition of flames and decreasing its thermal decomposition from 0.94 mg/min to 0.65 mg/min.

*4.2. Ecological Aspects*

Another prerequisite for applying wood surface treatments at a large scale lies in their environmental footprint. As the awareness of the protection of human health and the ecosystems is on the rise, there is a need for wood treatments to be continuously safer. A first step in this direction is the substitution of solvent-borne coatings and treatments with waterborne alternatives. Indeed, the volatile organic compounds (VOCs) contained in solvent-borne coatings have different consequences on the health of both humans, particularly for individuals with respiratory problems or high sensitivity to chemicals, and the environment [261]. This consideration can be regarded as a success, since only a few of the treatments presented in this review used organic solvents. On the contrary, waterborne coatings were extremely prominent, even though some additives or resins needed to be modified to improve their hydrosolubility.

A typical limitation of waterborne organic coatings, in comparison to their solvent-borne counterpart, is their high permeability to water. However, many coatings developed during the last few years could overcome this issue. For example, inorganic coatings relying on hydrothermal deposition do not require their precursors to possess a good affinity with water, as they can be deposited from a dispersion or grown from a redox reaction. Consequently, their permeability to water is highly reduced, even more so if they are subsequently modified with a low surface free energy compound. The deposition of highly hydrophobic coatings could also be achieved by plasma polymerization, where neither water nor organic solvents were needed.

Similar to plasma polymerization, many surface improvements could be achieved without a dispersant. The fluorination of wood surfaces achieved a greatly delayed absorption of water [215,216], particularly when combined with a short torrefaction [217]. Pouzet et al. found that reacting wood with gaseous $F_2$ for five minutes could substitute hydrophilic hydroxyls (-OH) for hydrophobic fluorines (-F), which increased the water contact angle on the modified wood, delayed its water absorption, and lowered its equilibrium moisture content. Moreover, the hydrophobicity of the so-treated timber barely decreased after two years. However, this method also involves the generation of HF as a by-product, which undermines the safety aspect. Methods relying on the carbonization of wood like $CO_2$ laser irradiation [225–227], one-sided charring [228–230] and blow-torch combustion [231] improved the resistance to biodegradation and the water sorption properties of the treated wood, although its appearance was notably affected (blackened) by the charring.

Another aspect of the safety of wood surface improvement is the leaching of the chemicals. A lot of the treatments presented in this review rely on silica, metal, and metal oxide nanoparticles, as well as other nano-sized materials. However, there is considerable concern about the release of these compounds since they can represent health hazards for both humans and the environment [262–264]. Additionally, they are expected to stay in the environment for a very long period as they are not readily biodegrade. The toxicity of nanometric compounds is quite complex, depending on many factors such as their ionic strength, size, surface properties, and aggregation. Moreover, they can react with their environment (soil components, moisture, acidity, etc.), which will change their properties, toxicity and mobility [265]. Nonetheless, nanoparticles can enter through the cell wall of mammalian cells and generate cytotoxic reactive oxygen species (ROS) responsible for DNA damages [266–268]. They were also found accumulated in various organs such as the lungs,

alimentary tract, liver, heart, spleen, and kidneys [269,270]. Adverse effects of nanomaterials were also noted in fish, where they could damage different tissues, create hormonal alterations and even become lethal at fairly low concentrations [271,272]. Although some benefits were observed in plants upon exposure to nanoparticles, different authors also noticed that they could affect the germination of seeds and the growth of the plants, as well as their quality and yield. Furthermore, their hormonal balances were disrupted and some stages such as the flowering and the fruiting could be delayed [273,274]. Fortuitously, many workers used bio-based nanomaterials to obtain similar properties with biodegradable compounds. Different cellulose nanomaterials were used as additives in coatings (film-forming or oils) to improve their mechanical properties; while nanocellulose is not completely safe either, its effect on health is far inferior, and it is readily biodegradable, unless it underwent too much modification [275–277]. As a result, bio-based systems with very high adhesion, hardness, tensile strength and resistance to wear were obtained [59,60,62,64,66–69]. Veigel et al. added nanofibrillated cellulose to linseed oil to improve its mechanical durability [65]. They found that the unmodified oil would quickly become less hydrophobic when exposed to abrasion, while the nanocellulose allowed it to repel water for a much longer time. Also, superhydrophobic coating was prepared by Huang et al. by spraying a nanofibrillated cellulose dispersion on wood specimens before its modification with 1H,1H,2H,2H-perfluorooctyltrichlorosilane [136]. In so doing, they created a wood surface with a 161° water contact angle, a roll-off angle <10°, self-cleaning, and extremely high durability, mimicking the typical protocol to obtain superhydrophobic coatings made with inorganic nanoparticles. Similarly, great photostability was obtained by substituting UV absorbers like ZnO and $TiO_2$ for wood extractives [70,71,73,74]. Grigsby added condensed tannins and modified tannins to acrylic coatings to compare their UV blocking capabilities to commercial hindered amine light stabilizers and phenolic stabilizers, finding that 0.5% of tannins could extend the coatings' life for longer than the commercial options [72]. A multifunctional coating was prepared by Huang et al. by adding cellulose nanofibrils with high lignin content along to western red cedar and lodgepole pine bark extracts in a water-based acrylic [75]. The coating they obtained showed both very good mechanical properties and less photodegradation when exposed to artificial aging.

In a more general fashion, many workers developed wood surface treatments containing bio-based materials over the last five years. These treatments, mostly belonging to the organic coatings' trends, included bio-based monomers for film-forming coatings [41–44,47], modified vegetal oils [39,40,45,46,51,55,56], biological polymers (chitosans, lignin, etc.) [57,58,92], and more [52,53,76,90,143–145]. Many advantages come along with these materials, from the cradle to the grave. In general, the activities to generate the raw materials for bio-based components are less energy intensive than their petroleum counterparts, which reduces the consumption of energy and unrenewable resources, as well as the production of greenhouse gases [278–281]. Since they originate from cleaner sources, they also produce less air pollution and they are safer for human health, the ozone layer and ecosystems. Conversely, bad decisions while managing crops destined to become bio-based chemicals can tarnish their health and ecological profiles [278]. The cultivation of plants destined for bio-refining is directly linked to massive land and fertilizer usage, the latter being responsible for eutrophication of surrounding water bodies. This drawback may, nonetheless, be avoided by promoting the valorization of wastes and residues from agricultural and forestry activities, when possible, which would diminish both the pollution and land use associated with the production of crops [282]. Bio-based materials also offer great opportunities in terms of eco-friendly disposal, as they are readily biodegradable just like wood [283,284]. Therefore, compostability may be regarded as an acceptable way to dispose of wood coated with bio-based coatings after its useful life. Additionally, bio-based materials are easier to incinerate than synthetic materials, consequently diminishing the recourse to landfilling [285]. The incineration of materials issued from photosynthesis, like wood and the other plants used to prepare the coatings, can be considered carbon neutral as the $CO_2$ produced by the combustion was previously sequestrated by the vegetal [286],

thereby decreasing its overall environmental impact. Furthermore, the heat produced by the incineration can be recovered to fuel different activities and avoid the consumption of unrenewable resources for energy production.

The implementation of passive systems is another great way to make wood surface treatments more ecological. The operational energy consumption of a building can be considerably reduced by using passive systems as compared to conventional systems [287]. For instance, the energy needed to heat and cool buildings, both residential and commercial, represent more than a third of their total energy consumption [288]. An approach studied over the last years to improve the energetic efficiency of buildings is the application of phase changing materials (PCMs) [289–292]. PCMs allow for the storage of thermal energy by melting the material and, subsequently, releasing this energy through solidification [293]. When using a material that melts near to room temperature (20–22 °C) in a building, it improves the thermal comfort of the habitants by sequestrating excessive heat, which diminishes the need for climatization. When the temperature falls below the melting point of the PCMs, the energy is released, reducing the need for heating. In a previous study, Mathis et al. demonstrated that composite walls containing PCMs could decrease the need for heating during the night by as much as 41% in timber-frame test huts simply by releasing heat accumulated during the day [294]. Two of the publications presented in this review detailed energy-storing coatings which contained PCMs. Kong et al. spray coated a wood substrate with a mixture of mesoporous polydivinylbenzene (PDVB) nanotubes, an industrial paraffin wax, and fluorine-containing $SiO_2$ nanoparticles to obtain a surface with both superhydrophobicity and thermal energy storage [107]. The PDVB nanotubes could absorb and retain a large amount of paraffin wax (78.29% in mass), leading to a latent fusion heat of 119.6 J/g. As a result, infrared thermography tests showed that the temperature of wood surfaces treated with the paraffin wax would change in a lesser extent than wood treated only with the nanotubes when exposed to heating (from a gain of 50 °C to 25 °C) and cooling (50 °C loss to 20 °C). However, the impact of the coating on the temperature of the room was not presented. Another functional coating was prepared by Qian et al. by encapsulating n-eicosane in $Fe_3O_4/SiO_2$ microcapsules through a sol-gel method, leading to a wood surface with thermal energy storage and magnetism [155]. This coating showed a promising heat storage capacity, with a melting enthalpy of 170.9 J/g; however, no test was performed to assess its effect on the temperature of the wood or the room. The development of coatings with new types of passive systems could be a great avenue for future research.

The authors believe that three publications presented in this review have set themselves apart for their novelty in terms of ecological wood surface treatments. First, a coating was developed by Janesch et al. that was not only entirely bio-based, but also food safe [51]. The coating, made of bee wax and tung oil, was dip-coated on spruce specimens before being sprinkled with NaCl. After a week-long drying, the salt could be removed with de-ionized water, leaving the organic layer with a well-defined micro-/nanoscale architecture. The so-casted coating was highly hydrophobic, with a water contact angle of 161°; however, it would not be considered superhydrophobic as its roll-off angle was way higher than 10°. Second, Zhang et al. prepared a jellified coating based on chitosan, gelatin and glycerol [52]. This coating had strong self-healing capabilities, being able to heal completely medium scratches simply by heating, and could easily be colored with water-soluble dyes. Its most innovative feature was its reusability, as it could simply be scraped from wood, dissolved in water and applied again. Finally, a silica-based coating with fire retardancy was prepared by Belykh et al. while using only industrial wastes [166]. The coating, which was made of by-products from the fabrication of ferrosilicon (sodium liquid glass) and gold mining activities (black shale), had a strong adhesion to the wood substrate and a lower mass loss after burning. While these treatments were not the best in terms of performances, they used unconventional approaches (food safety, reusability and wastes recycling), which should hopefully inspire future research.

### 4.3. Economical Aspects

The last criterion that wood surface technologies must satisfy in order to be applied in real-life applications that will be discussed in this review revolves around their industrialization. While do-it-yourself methods can sometimes be applied, most treatments presented in this review rely on materials and equipment that are not readily available to everyone. Consequently, their use become dependent of the economy and the dynamics affecting the producers and the consumers. In other words, the viability of a wood treatment becomes conditional on whether an industry is willing to offer the treatment and whether the consumers are willing to buy it.

An important aspect of this dynamic is the price. The competition for treated wood is quite fierce, as wood products compete amongst themselves as well as against other construction materials [5]. Consequently, a wood surface treatment must be cheap to produce, as otherwise either the consumer will not be interested to buy it (too expensive) or the producer will not be interested to sell it (no profit). Accordingly, treatment methods such as dipping, spraying, and painting are quite attractive as they do not require any expensive equipment, are quickly applied, and are adaptable to different substrate shapes. These methods were adopted in different publications presented in this review to improve the characteristics of the wood surfaces, such as mechanical properties, thermal stability, dimensional stability and resistance to biodegradation. They were particularly prevalent to cast inorganic [115–118,121,122,132,147,148,150,151,153,154,156–158,160–166,170,172] and organic-inorganic composite [87,88,103,105–107,109–114,125–127,129–131,134–137,142–144,146] coatings on different wood substrates. However, it is important to bear in mind that the application of these coatings can be lengthy; for instance, spraying and painting methods often require multiple layers, each of which must be dried before applying the next one. A similar pattern can be observed with dipping when building a layer-by-layer coating. Hydrothermal deposition was thoroughly studied to build different metallic coatings on wood; commonly, these treatments require multiple hours of dipping and even longer dryings. Likewise, sol-gels were extensively studied to build coatings based on $SiO_2$ and $TiO_2$, which also requires prolonged dipping periods, aging post-treatments and drying. Accordingly, these coating methods must allow for working in large batches in order to keep a good production rate. Brushing and dipping were also used to perform the surface impregnation of different protective agents into wood substrates, with similar limitations [238–241,244,246,247]. A quite different, yet interesting method was presented by Volokitin et al., who modified pine and birch with a thermal plasma to produce only a layer of thermo-modified wood [193]. The resulting timber displayed similar properties to thermo-modified wood (higher hydrophobicity, lower water absorption, dark colour), but was faster and cheaper to produce.

Some widely available and inexpensive chemicals were studied over the last five years as well. Among them, silica ($SiO_2$) received a particularly high attention as it showed great versatility in terms of properties, usage, and application methods. A first use for silica was as an additive in organic coatings to improve their mechanical properties [90,95] or to create a durable micro-/nanoscale architecture into the coating to reach superhydrophobicity [104]. Similar to the latter, it was also used to create a micro-/nanoscale architecture directly on the wood surface before hydrophobization with very low surface free energy compounds [104,106,111,113,117,118,120,126,128,130,132,133,151]. Although application methods such as dipping and spraying were typical, Yang et al. developed a technique to create a mold of the surface of plants with natural superhydrophobicity, such as canna leaves [139] and rose petals [140]. Once filled with a PVB/$SiO_2$ composite, the mold allowed for the deposition of a perfect replica of the original plant surface on the wood. A last use for silica was to create a solid layer into which other materials could be encrusted to imbue new properties, such as photocatalytic activity [153,154] and photostability [196]. Although most researchers acquired their silica from a commercial supplier, some workers studied more durable options such as rice husk [90] and industrial wastes [166]. Another cheap, mildly toxic [295], and vastly studied component was

TiO$_2$ [89,91,94,110,125,129,131,147,148,153,154,158,197,205,208]. Its uses were similar to silica's, in addition to which its UV absorption properties improved the photostability of the coated/treated wood. Finally, calcium carbonate (CaCO$_3$) was examined under different forms as an additive to organic coatings. When added to a waterborne acrylic resin as powdered oyster shell, it extended the burning time of the substrate from 18.00 min up to 29.67 min [97]. In order to prevent the premature degradation of organic UV absorbers in a clear acrylic coating, Queant et al. built CaCO$_3$ microcapsules to encapsulate and protect the UV absorbers, increasing their efficiency over a prolonged period [82].

Apart from using cheap methods and materials, another way to make wood treatments worthwhile would be through added value. While plasma treatments require expensive equipment [20], they are quite scalable and allow for the preparation of durable coatings. Over the last five years, many authors studied the utilization of non-thermal plasma treatments to increase the surface free energy of wood with the objective to improve its wettability and the penetration of coatings [177,180,181,186,188,189]. Haase et al. found that pre-treating black spruce with a glow-discharge plasma would improve the penetration of coatings without increasing their adhesion [176]. However, they noted that it would improve the performances of solvent-borne coatings after 3024 h of artificial weathering, showing a better aging behavior. On the other hand, Zigon et al. showed that the tensile strength of coatings applied on weathered wood could increase by 20% with a FE-DBD plasma pre-treatment [178]. These results were supported by Peng et al. who also noted that dielectric barrier discharge plasma treatments would consequently reduce the delamination of a water-based primer applied on a beech substrate during a cross-cut test [190]. Accordingly, by virtue of their ability to enhance the penetration and bonding of different coatings, plasma treatments could be particularly useful in the case of wood products that cannot readily be refreshed, such as flooring and cabinets. In these scenario, the extra cost may be worth the gain in product quality and durability, as Kozak et al. discovered that consumers would be willing to spend more money for a product with a proportionally higher quality [296]. Moreover, the preparation of more expensive, premium grade wood products could have many supplementary advantages, such as diversifying the offer of wood products with different grades of goods and avoiding giving timber products an image of solely low cost and grade materials. It is also safer to offer an interesting warranty on high quality products with less chances of defect [297], which is a powerful marketing tool with high regards from many customers expecting to use wood in their projects [298,299]. Apart from plasma treatments, another method to create high quality wood surfaces was the encapsulation of organic coating additives. Yan and Peng studied the effect of adding urea-formaldehyde microcapsules with an acrylic resin core to water-based paints, which showed that the microcapsules imbued the paint with a strong self-healing capacity even when numerous and large cracks would appear [81]. Similarly, Queant et al. encapsulated organic UV-absorbers in CaCO$_3$ microcapsules to reduce their premature degradation, which increased the photostability of a clear latex over 2500 h of artificial weathering [82]. Although the study of encapsulated coating fillers was fairly scarce over the last five years, the authors believe that it could be a promising route to develop new highly performant and durable wood surfaces.

In a more general fashion, the durability of wood surfaces and their properties is of crucial importance. No matter how performant a treatment is, it could hardly be considered a high-quality product if it loses its properties after just a while. The assessment of the most durable wood surface treatments was quite challenging for two main reasons. First, the methods used to test the properties of the wood surfaces, particularly their resistance to abrasion, were extremely variable, making comparison between the publications hazardous. Second, the durability tests, sometimes, were found to be unrepresentative of the stress the surfaces would encounter in their real condition of usage. As an example, sandpaper abrasion tests were extremely prominent for the durability of superhydrophobic surfaces, and often the only tested method. However, although such an abrasion may be representative for the surface of kitchen countertops or flooring, softer methods, such

as sand and drop-impact abrasion would be more representative of the degradation of fences, sidings, and claddings, which are more likely to host a superhydrophobic surface. This consideration is of major importance, as the publications using both sandpaper abrasion and a softer test showed that their treatment would lose their superhydrophobicity quite rapidly when subjected to sandpaper, while the softer method would only affect it marginally [114,121,123]. Consequently, a wood surface treatment that could be perfectly fit for a certain function may be deemed as non-durable simply because the testing method used was inadequate for this type of treatment. For this reason, the authors would advise researchers to verify that their testing method is the most appropriate one for the end-use of the wood surface protection they are developing, and to consider using multiple complementary methods if needed. As a general statement, coatings based on inorganic materials such as $SiO_2$, $TiO_2$, ZnO, and metals seemed like the most durable treatments, with high adhesion and resistance to mechanical (scratches, abrasion, and cutting) and chemical (acids, alkali and organic solvents) deterioration. Similar components and nanocellulose successfully acted as additives in organic coatings to improve their mechanical properties as well [59,60,65,69,75,95,96]. From all the different surface properties presented in this review, superhydrophobicity was the most tested for its performance following degradation. Although most superhydrophobic treatments lost their superhydrophobicity upon exposure to degradation, some of them stood out, such as Ou et al.'s silane composite coating, which would remain superhydrophobic or nearly superhydrophobic even after abrasion (sandpaper or tape peeling), knife scratching, immersion in ethanol, acids, and alkali, and exposition to UV radiations [115]. Similarly, Wang et al. prepared a composite coating with polydopamine, copper, and octadecylamine that would remain superhydrophobic after exposition to UV radiation or 24 h of soaking in HCl, NaOH, different organic solvents and boiling water [135]. A solution to the generally poor durability of superhydrophobic coating may be found in self-healing, as the restoration of the micro-/nanoscale architecture following mechanical damage can revert the loss of hydrophobicity [21]. Accordingly, different workers developed superhydrophobic coatings with self-healing capacities over the last five years, although some of them require a thermal stimulus for the actual healing to take place [110,116]. Conversely, a self-healing coating prepared by Wang et al. with $SiO_2$ and perfluorooctyltriethoxysilane could not only self-repair upon taking damage without needing any external stimulus, but its self-healing ability could be replenished by spraying a coating solution [133]. Additionally, since superhydrophobicity (contact angle > 150°) offers arguably greater performance than actually necessary in practice [124], another solution would be to accept a lower contact angle to improve the durability of the wood surface. Since superhydrophobicity requires a quite strict and fragile architecture at the nanometric level, a rougher surface with low energy could allow for a satisfying hydrophobicity while being more resistant to damages. The durability of a flame-retardant system surface impregnated in poplar wood was tested by He et al. after a leaching experiment [241]. This system, establishing the synergistic effect of $SiO_2$ and $K_2CO_3$ on the limiting oxygen index (LOI) of the treated poplar, allowed for the obtention of a satisfying 30.5% LOI even after leaching. Finally, wood with very high impact resistance was prepared by surface impregnating reactive chemicals prior to one-sided surface densification [232–234].

The visual aspect of wood is also a crucial element for wood surface treatments. Different studies have shown that the appearance of wood is of foremost importance to the consumer [300,301], and the natural aspect of wood is usually well appreciated [302,303]. In regard to wood surface treatments, they represent an extra challenge to efficiently protect wood, as transparent surfaces cannot block UV radiations, which can lead to the photodegradation of the wood polymeric constituents and the delamination of clear coatings [24]. Satisfyingly, many publications over the last five years undertook to solve this problem by different means. Inorganic coatings made of different metallic oxides, mainly ZnO, demonstrated great abilities to prevent color changes and chemical degradation in wood [122,147–149,154,156,160,161]. Similarly, organic-inorganic composite coatings [105,109] and plasma-deposited coatings [195,204,205,207,208] also used metallic ox-

ides to achieve comparable performances. Clear organic coatings could be imbued with UV absorption when using different additives, among which wood extractives were particularly studied [70,71,73–75,82,94]. Grigsby discovered that a loading of less than 0.5% of condensed tannins and modified tannins in an acrylic coating could extend its life for longer than commercial HALS and phenolic stabilizers during accelerated and natural weathering [72]. However, the tannins would also make the coating darker. Indeed, another challenge associated with transparent wood treatments is for the treatment to be as clear as possible itself in order to preserve the natural color and gloss of pristine wood. Some workers noticed that their wood surface treatment would modify the appearance of wood [44,59,93,96,171,206,225,226,230]. However, it was very uncommon for authors to present the effect of their treatment on the transparency of wood although it is a very relevant data. This is particularly important when wood is not expected to be exposed to sun and undergo color changes, as the initial aspect modification will not be attenuated over time. Conversely, as stated by Sedliaciková [302], a significant proportion of the public is in fact looking for colored wood. While dyes and pigments are commonly used for this purpose, innovative new methods were developed over the last few years to obtain colored wood surfaces. Dagher et al. surface-impregnated a ferric sulfate solution in various hardwood species to produce colored complexes with their phenolic extractives, generating new colors directly inside of the wood [237]. Also, polystyrene colloidal microspheres with different acrylate-based copolymers were prepared by Liu and Hu to obtain organic coatings with colorful hues of green, red, and orange. Finally, many authors presented different transparent systems that could change color under stimuli such as UV radiations [161–163] and heat [77–80,242].

As a general statement, coatings based on inorganic materials performed very well from an economic perspective, as many of them used cheap materials such as $SiO_2$ and $TiO_2$, simple methods, such as dipping and painting, and possessed a very high durability. An interesting example of such coating was provided by Wang et al. who produced a liquid-like $SiO_2$-g-PDMS coating with a contact angle of 91° [120]. Although this coating was far less hydrophobic than most coatings described in this review, it had the particularity to remain hydrophobic after prolonged contact with water. While most coatings undergo a transition from the Cassie−Baxter state to the Wenzel state after being in contact with water [304], which means that the air trapped in the micro-/nanoscale architecture is lost and that water can now adhere freely to the wood surface, this coating would keep a stable water contact angle even after 19d of immersion. This feature could be very interesting to protect wood exposed outdoor in regions affected by heavy rains and in other circumstances where contact with water is extremely frequent. A very promising wood modification method was developed by Pouzet et al. to substitute the many hydroxyl groups at the surface of wood with hydrophobic fluorines ($F^-$) by using gaseous $F_2$ [215,216]. The resulting wood surface displayed a high water contact angle (120°), a slower water absorption, and a very high durability as its performances remained unchanged after two years. Furthermore, it did not affect the color of Douglas fir and the water absorption could be further reduced by torrefaction [217]. Although the method relies on more expensive equipment and requires a rigorous sample preparation, its very short treatment time (5 min), scalability, low impact on the substrate appearance, the low cost of the chemicals involved, and its overall performances and durability shows great potential for the future.

The characteristics discussed in the last few pages are summarized in Table 3 This table only represents a summary of the aspects mentioned in the discussion, and should not be considered as the complete list of the characteristics for these trends. To conclude this discussion, this last paragraph will mention subjects the authors believe deserved more attention. First, the encapsulation of active compounds in stimuli-responsive microcapsules could represent a great way to increase the longevity of the properties of coatings. By protecting the active ingredients from external sources of degradation such as moisture, UV radiations, micro-organisms, and heat, properties such as photostability, fire-retardation, and fungal resistance could be extended to produce higher quality wood surfaces. Also,

while protection from liquid water received the most attention of all the properties presented in this review, only a few authors presented results in regard to moisture buffering. Yet, this aspect is of major importance, as relative humidity alone can promote the growth of decay fungi and produce consequent dimensional changes in wood. Finally, none of the publications presented in this text presented the freeze-thaw properties of their wood surface treatment. However, an important fraction of the world population resides in areas where winter temperatures can reach below the freezing point. Consequently, this aspect could be a determinant for the longevity of exterior coatings.

**Table 3.** Characteristics of the different trends which were presented in the discussion.

| Categories | Trends | Characteristics |
|---|---|---|
| Coatings | Organic coatings–Reactives | Creation of interesting fire-retardant coatings. |
| | Organic coatings–Bio-based reactives | Great potential to replace less environment-friendly, oil-based monomers when using the rights sources of raw materials; protection against biodegradation. |
| | Organic coatings-Other | Can develop original properties such as sea-water resistance and bright colors. |
| | Organic coating additives-Bio-based | Greater durability through increase mechanical resistance and photostability; biodegradable upon leaching. |
| | Organic coating additives–Non-biobased | Wide variety of properties such as mechanical resistance, fire-retardation, energy storage, and thermochromism; allows the encapsulation of functional materials. |
| | Organic coating additives–Metallic oxide nanoparticles | Important properties such as mechanical resistance, hydrophobicity, and photostability; cheap materials, but hazardous upon leaching. |
| | Organic coating additives–Minerals | Important properties such as mechanical resistance and hydrophobicity. |
| | Organic-inorganic composites–Nanoparticles + organic resin | Important properties such as mechanical resistance, hydrophobicity, moisture buffering, and photostability; cheap materials and methods, but hazardous upon leaching. |
| | Organic + inorganic composites–Nanoparticles + low surface energy components | Important properties such as mechanical resistance, fire-retardancy, hydrophobicity, moisture buffering, and photostability, good potential for self-healing; cheap materials and methods, but hazardous upon leaching. |
| | Organic + inorganic composites–Others | Important properties such as fire-retardancy and hydrophobicity. |
| | Inorganic coatings–Metallic oxides and silica | Important properties such as mechanical resistance, fire-retardancy, hydrophobicity, moisture buffering, and photostability, good potential for self-healing; cheap materials and methods, but hazardous upon leaching. |
| | Inorganic coatings–Others | Important properties such as mechanical resistance, fire-retardancy, hydrophobicity, and photostability; interesting properties, e.g., magnetism. |
| Surface modification | Plasma–Hydrophilization | Expensive, but produces high quality timber products with improved coatings adhesion and durability. |
| | Plasma–Coating deposition | Important properties such as hydrophobicity and photostability; expensive, but solvent-free. |
| | Other modifications–Chemical modification | Important properties such as hydrophobicity and moisture buffering; no or low leaching, can be solvent-free. |
| | Other modifications–Carbonization | Solvent-free methods to improve the durability against biodegradation, but heavily affects the color of wood. |
| Surface impregnation | Impregnation of reactives | Improvement of the hardness of wood surfaces; development of colors. |
| | Impregnation of material | Important properties such as fire-retardancy and hydrophobicity; lower WPG than pressure impregnation. |

## 5. Conclusions

Increasing the use of wood in construction is a great way to combat climate change while developing an aesthetic and durable building stock. In order to achieve this goal, work must be undertaken to convince the public and architects to integrate more wood in buildings. Scientists can assist in this task by making wood more appealing, either by

protecting it from hazards such as combustion, biodegradation and photodegradation, or developing new properties, e.g., self-healing and self-cleaning.

Over the last five years, at least 212 publications were written to describe innovative methods to improve the surface of wood. In this review, these publications were divided into different trends based on the chemical composition and the treatment method they described. After introducing the different trends and the publications that comprise them, various normative, ecological, and economical challenges associated with the protection of wood were presented. The way different trends could answer to these challenges was then discussed, leading to a certain hierarchy for potential real-life applications. Each section was finally concluded by describing the individual treatments that showed the most originality or performance toward the discussed subject.

The discussed aspects of wood protection showed that trends including inorganic nanoparticles such as silica and metallic oxides showed an incredible potential for different applications. They could first answer to normative principles by granting fire-retardancy to the protected wood and reducing the hazards of biodegradation through a drastic diminution of the uptake of water. From an economical point of view, they use cheap and abundant materials, are applicable with easy methods, are highly durable, and can efficiently protect the wood polymeric constituent from UV radiations. However, the leaching of these nanoparticles into the environment may represent a serious hazard in the long run, as they possess a certain toxicity and cannot be naturally degraded, which will lead only to increasing concentrations.

This review should help researchers to plan their future projects with a broader perspective of the different criteria that wood surface improvements need to meet in order to be introduced in the society. By focusing on the development of cheap, durable, and environment-friendly treatments to solve issues such as combustibility, photodegradation, and biodegradation, a new generation of wood surface technology could revolutionize the perception of the public toward wood. The authors hope that the reflections shared in the review will help to accelerate the transition of new and innovative wood treatments from the laboratory to the market so that they can contribute to increase the use of wood in buildings.

**Author Contributions:** Structure of the work, P.B.; Review of the literature and writing of the original draft, S.P.; review and final editing of the paper, S.P. and P.B. All authors have read and agreed to the published version of the manuscript.

**Funding:** This research was funded by Natural Sciences and Engineering Research Council of Canada, grant number IRCPJ 461745- 18 and RDCPJ 524504-18.

**Institutional Review Board Statement:** Not applicable.

**Informed Consent Statement:** Not applicable.

**Data Availability Statement:** Not applicable.

**Acknowledgments:** The authors are grateful to the Natural Sciences and Engineering Research Council of Canada as well as the industrial partners of the NSERC industrial chair on eco-responsible wood construction (CIRCERB).

**Conflicts of Interest:** The authors declare no conflict of interest.

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
