# Peer review of "Trends in Chemical Wood Surface Improvements and Modifications: A Review of the Last Five Years"

_coatings, doi:10.3390/coatings11121514_

Round 1

Reviewer 1 Report

The idea of the review article is very interesting.

Theatrically the paper is very sound, but the article lack of some additional Tables and Figures that show the most used coating materials and their effects on wood or wood products. On other meaning, why there are no Table showing the coating materials used,  the wood type ,,  wood products, application method, their effects on wood structure, …..

Author Response

Dear reviewer,

Thank you for reading our review and for sharing your comments.

R1: Theatrically the paper is very sound, but the article lack of some additional Tables and Figures that show the most used coating materials and their effects on wood or wood products. On other meaning, why there are no Table showing the coating materials used,  the wood type ,,  wood products, application method, their effects on wood structure, …..

A: Because the review presents more than 200 publications using a large combination of chemicals, application methods, and wood species, such tables would nearly need to present each treatment individually. Consequently, they would greatly increase the length of an already very long publication. Moreover, since most wood surface treatments are already presented twice, in the sections 3 and 4, the authors believe that additionally presenting them in tables would lead to redundancy. However, the main features from the different treatments were presented in Table 3 after sorting them in trends.

The authors believe that a review should focus on summarizing only the most relevant informations of the presented publications in order to guide the readers to relevant publications. Accordingly, as mentioned on line 208, the readers are greatly encouraged to read the original papers if they would like to learn more about the details which could not be included in the review. We are sorry that the revision did not lead to more changes and hope that our explaination could diminish your concerns.

Reviewer 2 Report

With regard to the above mentioned manuscript, the topic is interesting and well-suited for your esteemed journal. The whole manuscript is clearly written and there is a logical flow of information throughout the text. However, there are some concerns which I recommend being reconsidered by the authors as follows:

  • Introduction: My primary concern is this section. While I approve all that is already written and mentioned in this section, I believe there is something of great importance missing. Over the past few years or so, addition of some mineral materials containing silicon compounds (particularly wollastonite) to wood and wood-based composite mats (particleboard and MDF panels) were investigated from different perspectives. I noticed that none of these published papers were cited in this manuscript. The results of these papers can greatly help the Discussion section of the manuscript, giving reasons why the results were achieved and why water repellency in the present panels was improved. Moreover, effects of nano-silver on paint pull-off strength is also of great importance as paints cover the surfaces of wood and wood-based panels, and any alteration in the quality of the surface would eventually affect the strength of paints on them. Therefore, I suggest the authors elaborate more and find some new studies in which wollastonite (at nano- or micro-scales) was added to resins or paints. In this regard, studies conducted by Prof. Antonios Papadopoulos, Prof. Ayoub Esmailpour, Prof. Jeff Morrell, Prof. Petar Antov, Prof. George Mantanis, and Dr. Roya Majidi are highly recommended. Ignoring these studies would not only decrease the quality of the manuscript, but also the lack of reasoning and deductions that the above mentioned authors have provided in their studies would make it very difficult for the present authors to provide an agreeable discussion as to the reasons of improvement in the repellency of the panels produced.
  • Another topic that I think was a bit ignored is “Smart Windows” and “Transparent Wood”. These two topics are closely related to the way light is past through or reflected by the surface of woody members in a structure. Therefore, addition of these two topics, though very briefly, would add to the quality of the paper, and substantially increase future readers and citation to the published paper. In this regard, studies carried out by the following researchers would greatly help: Prof. Deb, Prof. Banerjee, Prof. DeForest, Wang, Dr. Li, Prof. Zhu, and Runnerstorm.

Author Response

Dear reviewer,

Thank you for reviewing our text and for sharing your point of you, we which lead to the following modifications:

R2: Introduction: My primary concern is this section. While I approve all that is already written and mentioned in this section, I believe there is something of great importance missing. Over the past few years or so, addition of some mineral materials containing silicon compounds (particularly wollastonite) to wood and wood-based composite mats (particleboard and MDF panels) were investigated from different perspectives. I noticed that none of these published papers were cited in this manuscript. The results of these papers can greatly help the Discussion section of the manuscript, giving reasons why the results were achieved and why water repellency in the present panels was improved. Moreover, effects of nano-silver on paint pull-off strength is also of great importance as paints cover the surfaces of wood and wood-based panels, and any alteration in the quality of the surface would eventually affect the strength of paints on them. Therefore, I suggest the authors elaborate more and find some new studies in which wollastonite (at nano- or micro-scales) was added to resins or paints. In this regard, studies conducted by Prof. Antonios Papadopoulos, Prof. Ayoub Esmailpour, Prof. Jeff Morrell, Prof. Petar Antov, Prof. George Mantanis, and Dr. Roya Majidi are highly recommended. Ignoring these studies would not only decrease the quality of the manuscript, but also the lack of reasoning and deductions that the above mentioned authors have provided in their studies would make it very difficult for the present authors to provide an agreeable discussion as to the reasons of improvement in the repellency of the panels produced.

A: Thank you for bringing these researches to our attention. After investigating the mentioned subjects and authors, we found that the proposed treatments were either applied by vacuum impregnation or incorporated to composite wood materials, which are both excluded from this review. Consequently, we could not add these publications to our work, as it would require to include two vast wood treatment domains and would make this review way too long . We however realized that this exclusion was not as clearly mentioned as intended in the methodology and hopefully rephrased it in a manner to avoid future misunderstandings (line 134). While they did not include these subjects in the review, the authors believe that a new review covering these subjects and other publications rejected by our exclusion criterions would indeed be of high interest.

R2: Another topic that I think was a bit ignored is “Smart Windows” and “Transparent Wood”. These two topics are closely related to the way light is past through or reflected by the surface of woody members in a structure. Therefore, addition of these two topics, though very briefly, would add to the quality of the paper, and substantially increase future readers and citation to the published paper. In this regard, studies carried out by the following researchers would greatly help: Prof. Deb, Prof. Banerjee, Prof. DeForest, Wang, Dr. Li, Prof. Zhu, and Runnerstorm.

A: The presentation of transparent wood would be outside of the scope of the review, as the wood template must be completely impregnated twice during the process (lignin removal/modification and resin impregnation) while the review focuses on surface treatments.  Moreover, the high amount of resin into the wood makes it rather a composite material, which also does not fit into our scope.

We hope that although our changes were fairly slim, they could correctly address your concerns.

Reviewer 3 Report

This work is of high scientific value and very useful for people searching for fundamental as well as practical knowledge in the field of wood surface protection.

Nevertheless, I think that some changes need to be made for the manuscript to be ready for publication.

The most important drawback of the manuscript is that, even though the authors claim that they reviewed surface modification methods, they partly present current knowledge and subjectively exclude an important part. A very good example is surface densification. The authors have excluded densification without considering surface densification technologies while on the same time they have included surface impregnation.

They somehow make a quick reference about Thermomechanical densification of wood without considering the great wealth of related works in this field (mostly presented in wood-oriented journals). They also do not mention combination of different thermomechanical (THM) or chemical (impregnation densification) methods.

Another example of poor review is that Thermal treatments were considered not relevant, but the authors refer to carbonization in the “other surface treatment” methods. Is carbonization not a result of thermal modification?

My suggestion is to revise the title and make it somehow more specific to the subjects covered by the manuscript. If the authors choose not to change the title, they need to present more information about other surface densification technologies and probably others too. They should probably try to find more references about these technologies (probably by using other databases)

Author Response

Dear reviewer,

Thank you for taking the time to review our work and for helping us to improve its content. Here are our answers to your comments:

R3: The most important drawback of the manuscript is that, even though the authors claim that they reviewed surface modification methods, they partly present current knowledge and subjectively exclude an important part. A very good example is surface densification. The authors have excluded densification without considering surface densification technologies while on the same time they have included surface impregnation.

A: Although the authors are aware of the great amount of publications describing the modification of wood through physical and thermomechanical means, these treatments usually affect a deeper part of the wood surface than what was meant to be presented in this review. Moreover, considering the length of the review and the amount of publications of physical wood treatments, we believe that adding surface densification methods would make this text too long while surface densification would be well worthy of a separate review on its own.

R3: They somehow make a quick reference about Thermomechanical densification of wood without considering the great wealth of related works in this field (mostly presented in wood-oriented journals). They also do not mention combination of different thermomechanical (THM) or chemical (impregnation densification) methods.

A: The thermomechanical densification was not included into the review as the compression leads to a large reduction of the timber’s thickness, affecting more than the outter layer of wooden cells. However, we did include the surface impregnation of reactive materials which could reduce the set-recovery following the compression. The exclusion criterion for this kind of wood treatments was hopefully rephrased in a clearer way on line 135.

R3: Another example of poor review is that Thermal treatments were considered not relevant, but the authors refer to carbonization in the “other surface treatment” methods. Is carbonization not a result of thermal modification?

A: Thermal treatments were considered out of the scope of the review as they affect the treated wood up to the core. On the other hand, the publication by Volokitin et al. [193], which presents a thermal plasma treatment of the wood surface resulting in properties similar to those thermally-modified wood, was presented in the review as its modification is only superficial. Likewise, the carbonization methods presented in the review affect only the surface of the treated timber, explaining their inclusion.

R3: My suggestion is to revise the title and make it somehow more specific to the subjects covered by the manuscript. If the authors choose not to change the title, they need to present more information about other surface densification technologies and probably others too. They should probably try to find more references about these technologies (probably by using other databases)

A: Thank you for your remark, as the title indeed had a wider meaning than intended for our already important work of review, we changed it to "Trends in chemical wood surface improvements and modifications: A review of the last five years" to limit its scope to chemical treatments.

Thank you again for reviewing our work, and we hope that our changes and clarifications could adress all of your concerns.

Round 2

Reviewer 1 Report

The authors didn't answer my comments well, or they didn't need to make some efforts to amend the article according to my comments.

Author Response

All the concerns have been checked by editorial office and senior editors.
